# Synergy over Discrepancy: A Partition-Based Approach to Multi-Domain LLM Fine-Tuning

**Hua Ye[1,2], Siyuan Chen[3], Haoliang Zhang[4], Weihao Luo[5], Yanbin Li[6], Xuan Zhang[2,7†]**

[1]Nanjing University    [2]Airon Technology CO., LTD    [3]University of Bristol
[4]The University of Oklahoma    [5]Donghua University
[6]Beijing University of Posts and Telecommunications    [7]Carnegie Mellon University

## Abstract

Large language models (LLMs) demonstrate impressive generalization abilities, yet adapting them effectively across multiple heterogeneous domains remains challenging due to inter-domain interference. To overcome this challenge, we propose a partition-based multi-stage fine-tuning framework designed to exploit inter-domain synergies while minimizing negative transfer. Our approach strategically partitions domains into subsets (stages) by balancing domain discrepancy, synergy, and model capacity constraints. We theoretically analyze the proposed framework and derive novel generalization bounds that justify our partitioning strategy. Extensive empirical evaluations on various language understanding tasks show that our method consistently outperforms state-of-the-art baselines.

## 1  Introduction

Large language models (LLMs) have propelled natural language processing (NLP) to unprecedented capabilities [Kumar, 2024, Karanikolas et al., 2023, Hu et al., 2025, Zhang et al., 2025b, Lin et al., 2025], owing largely to their extensive pretraining on massive, diverse textual corpora [Wu et al., 2022, Li et al., 2024a, Chen et al., 2025b, Zhang et al., 2024a, Sun et al., 2025b]. Fine-tuning these pretrained models for a specific downstream domain has been widely explored and proven highly effective; notable approaches include adapter-based modules [Houlsby et al., 2019, Zhang et al., 2024b], parameter-efficient fine-tuning via low-rank updates [Hu et al., 2021], and instruction-based fine-tuning methods [Cao et al., 2024]. These methods have also accelerated the application of LLMs in many scenarios, such as healthcare[Tong et al., 2025, Liu et al., 2025, Wang et al., 2025], transportation[Yao et al., 2023, Lu et al., 2025b, Zeng et al., 2025b,a], robotics[Xiao et al., 2025a, Yan et al., 2025, Zhang et al., 2025a], and other applications [Sun et al., 2025a, Zhang et al., 2025c].

However, while these methods excel in adapting to a single domain, practical scenarios frequently require simultaneous adaptation to multiple distinct domains-a scenario far less studied and substantially more challenging [Lu et al., 2025a, Chen et al., 2025a]. Due to constraints on energy or computing power [He, 2025], we must consider how to enhance the model's ability to adapt to different domains. Consider, for instance, a scenario where a single pretrained model must be simultaneously adapted to clinical texts [Thirunavukarasu et al., 2023], social media posts [Yang et al., 2024a, Jiang et al., 2024, 2025], and legal documents [Seabra et al., 2024]. Naive approaches such as jointly fine-tuning the model across all domains or independently fine-tuning separate models per domain often yield suboptimal results[Xiao et al., 2025b, Zhang et al., 2023, Wang and Zhang, 2024, Wen et al., 2023]: domain-specific features may negatively interfere, causing one domain to overshadow others or impair overall generalization capabilities [Lu et al., 2024a, Zheng et al., 2024, Van Veen et al., 2023, Tan et al., 2025]. This motivates our central research question: *how can we*

---

†Corresponding author(xuanzhang2199@gmail.com)

*effectively and efficiently fine-tune a single LLM across multiple heterogeneous domains, exploiting inter-domain synergies while mitigating negative interference?*

A natural strategy to mitigate these challenges is to add adapter modules specialized for each domain [Houlsby et al., 2019, Zhang et al., 2024b, Tao et al., 2023]. While adapters reduce the need for full fine-tuning, they do not fully address the complexity of managing multiple source domains with potentially large discrepancies. In such cases, the learned model has to juggle both shared features (common linguistic properties across domains) and domain-specific features (rare words, styles, or content) [Lu et al., 2022]. Existing approaches often handle these competing demands by imposing either domain adversarial objectives [Ganin and Lempitsky, 2015], distribution alignment [Peng et al., 2019], or low-rank parameter updates [Hu et al., 2021]. However, these techniques may still fail to exploit synergistic relationships-where certain domains are complementary and can reinforce each other's accuracy-and do not fully capture how best to "partition" domains to avoid negative interference.

Motivated by this gap, we propose a partition-based multi-stage framework for multi-domain LLM fine-tuning. Instead of jointly or separately adapting to all domains, our method clusters synergistic domains while isolating highly distinct ones. Our approach integrates practical constraints-memory budgets, domain shifts, and synergy opportunities-with theoretical insights on generalization benefits from restricted updates and strategic domain grouping. The main contributions are:

1) We introduce a novel partitioning algorithm that clusters domains according to their synergy and discrepancy, then fine-tunes the LLM in multiple stages. This orchestrated process prevents cross-domain contamination while leveraging beneficial interactions.

2) We derive new bounds that capture domain discrepancy, synergy offsets, and adapter complexity, establishing conditions under which multi-stage partitioning yields provably tighter guarantees than single-stage or naive multi-domain methods.

3) Our experiments demonstrate that partition-based multi-stage fine-tuning outperforms state-of-the-art baselines, improving accuracy across different domains and tasks while reducing memory usage.

## 2 Related Works

### 2.1 LLM Fine-Tuning

The success of LLMs such as GPT-3 [Brown et al., 2020], LLaMA [Touvron et al., 2023], and Falcon [Almazrouei et al., 2023] has underscored the importance of *efficient* fine-tuning. Fully updating model parameters incurs high computational costs [Zhang et al., 2024b, Hu et al., 2021], prompting parameter-efficient methods that modify only a small subset of parameters. Examples include adapter modules [Houlsby et al., 2019], low-rank projections (LoRA) [Hu et al., 2021], and tightly integrated adapters for LLaMA [Zhang et al., 2024b]. However, most methods assume a single domain; adapting LLMs efficiently to multiple domains remains challenging. Another related field is continual learning [Xu et al., 2025], but it assumes a sequential arrival of tasks, which differs from the setting in this paper.

### 2.2 Multi-Domain Data Learning

Real-world data often originate from disparate sources with distinct distributions and vocabularies [Ganin and Lempitsky, 2015, Sener and Koltun, 2018, Li et al., 2023, Zhang et al., 2024c, Chen et al., 2024, Xiao and Liu, 2025, Ke et al., 2025]. Traditional multi-domain methods align representations through adversarial training [Pei et al., 2018, Ganin and Lempitsky, 2015], moment matching [Peng et al., 2019], or multi-task objectives [Royer et al., 2024, Shen and Zhang, 2025], yet typically neglect inter-domain synergies. Adapting LLMs further complicates this via memory overhead, forgetting pretrained knowledge, and cross-domain contamination. Adapter-based solutions partially address these concerns [Houlsby et al., 2019] but rarely exploit domain partitioning to maximize synergy. Our partition-based multi-stage approach systematically clusters domains to leverage synergy and minimize discrepancy, providing theoretical guarantees.

## 2.3 LLM Data Selection

The efficacy of supervised fine-tuning (SFT) heavily depends on the quality and composition of training data[Yao et al., 2024]. Recent advances have introduced diverse metrics for data selection: *instance-level* criteria like perplexity [Cao et al., 2024], reward scores [Gou and Nguyen, 2024], and loss disparities [Li et al., 2024b], trajectory-based clustering via small proxy models [Yang et al., 2024b], as well as token-level selection methods [Lin et al., 2024]. Existing works largely overlook one critical factor: *domain interactions*, where diversity metrics operate at instance/token levels [Pang et al., 2024] without modeling cross-domain compatibility [Li et al., 2025b]. Our approach addresses this gap by systematically clustering domains for joint tuning.

# 3 Theoretical Analysis

## 3.1 Preliminaries and Notation

Let us consider $k$ distinct source domains $\{\mathcal{D}_1, \ldots, \mathcal{D}_k\}$, each containing samples $(x, y)$ drawn from some distribution over $\mathcal{X} \times \mathcal{Y}$. We have a large language model $\mathrm{LLM}_{\theta^*}$ pretrained on a massive corpus $\mathcal{D}_{\mathrm{pretrain}}$, and we assume $\theta^* \in \mathbb{R}^p$ lies in a high-dimensional parameter space. Our goal is to *adapt* $\theta^*$ (often minimally) to each domain $\mathcal{D}_j$ by introducing or modifying a small set of parameters (e.g., adapter modules) denoted by $\phi_j \in \mathbb{R}^{q_j}$, where $q_j \ll p$.

**Definition 1** (Multi-Source Fine-Tuned Model). A **multi-source fine-tuned model** is given by

$$f_{\theta, \{\phi_j\}}(x) = \mathrm{LLM}_{\theta^* + \Delta\theta}\Big(\phi_1, \ldots, \phi_k\Big)(x), \tag{1}$$

where $\Delta\theta \in \mathbb{R}^p$ is a (potentially small) update to the pretrained backbone $\theta^*$. Each $\phi_j$ may represent additional parameters specialized to domain $j$. The model internally selects or combines relevant $\phi_j$ based on domain context or training strategy.

**Loss and Risk.** We let $\ell\big(f(x), y\big)$ be a nonnegative loss function (e.g., cross-entropy) measuring the prediction error on a sample $(x, y)$. For a single domain $\mathcal{D}_j$, the *expected risk* is

$$\mathcal{L}(\theta, \{\phi_j\}; \mathcal{D}_j) = \mathbb{E}_{(x,y)\sim\mathcal{D}_j}\Big[\ell\big(f_{\theta, \{\phi_i\}}(x), y\big)\Big]. \tag{2}$$

When training on multiple domains jointly, we typically minimize an aggregated objective:

$$\mathcal{L}_{\mathrm{agg}}(\theta, \{\phi_j\}) = \sum_{j=1}^{k} \alpha_j \, \mathcal{L}\big(\theta, \{\phi_i\}; \mathcal{D}_j\big), \tag{3}$$

where $\alpha_j \geq 0$ with $\sum_j \alpha_j = 1$. If $\alpha_j = \frac{1}{k}$ for all $j$, we obtain a simple average risk.

## 3.2 Assumptions and Domain Discrepancies

To handle multi-source adaptation rigorously, we introduce assumptions on data distributions, smoothness, and domain overlaps.

**Assumption 3.1** (Lipschitz Loss and Smoothness). Assume $\ell(\hat{y}, y)$ is $L$-Lipschitz in $\hat{y}$. Moreover, suppose for any $(\theta, \{\phi_j\})$ and $(\theta', \{\phi_j'\})$, the difference in model outputs is bounded by a constant factor in terms of $\|\theta - \theta'\|_2$ and $\|\phi_j - \phi_j'\|_2$. Formally, there exists a constant $B > 0$ such that

$$\big\|f_{\theta, \{\phi_j\}}(x) - f_{\theta', \{\phi_j'\}}(x)\big\| \leq B\Big(\|\theta - \theta'\|_2 + \sum_{j=1}^{k} \|\phi_j - \phi_j'\|_2\Big). \tag{4}$$

**Definition 2** (Domain Discrepancy). Let $d(\mathcal{D}_i, \mathcal{D}_j)$ be the $\mathcal{H}\Delta\mathcal{H}$-*distance* between two domain distributions $\mathcal{D}_i$ and $\mathcal{D}_j$. Concretely,

$$d(\mathcal{D}_i, \mathcal{D}_j) = \sup_{h \in \mathcal{H}} \Big| \Pr_{x \sim \mathcal{D}_i}\big[h(x) = 1\big] - \Pr_{x \sim \mathcal{D}_j}\big[h(x) = 1\big] \Big|, \tag{5}$$

where $\mathcal{H}$ is a suitable hypothesis class.

### 3.3 Complexity of Multi-Source Adapter Updates

To preserve the implicit regularization from pretraining (i.e., the beneficial "low-complexity" region $\theta^*$ has converged to), one typically *restricts* either $\Delta\theta$ or $\{\phi_j\}$ or both. We quantify this through norms/penalties:

**Assumption 3.2** (Restricted Adapter Complexity). There exist constants $\rho_\theta, \rho_\phi > 0$ such that

$$\|\Delta\theta\|_2 \ \leq \ \rho_\theta, \quad \|\phi_j\|_2 \ \leq \ \rho_\phi \quad \forall j \in \{1, \ldots, k\}. \tag{6}$$

If $\rho_\theta$ is very small (or zero), this means the backbone remains near $\theta^*$; if $\rho_\phi$ is small, each domain adapter is limited in capacity.

Consider a Transformer-based large language model (LLM) with $L$ layers, each layer containing a multi-head attention (MHA) sub-layer and a feed-forward network (FFN) sub-layer, plus optional adapter modules for each of $k$ domains. Suppose each attention weight matrix $W_{\text{attn}}^{(\ell)}$ and feed-forward matrix $W_{\text{ffn}}^{(\ell)}$ is constrained by a spectral norm bound (or operator norm) $\|W\|_\sigma \leq \Omega_{\text{core}}$. Each domain adapter $\phi_j^{(\ell)}$ at layer $\ell$ is constrained by $\|\phi_j^{(\ell)}\|_F \leq \Omega_{\text{adapt}}$. We assume a bounded input embedding norm $\|x\| \leq C_{\text{in}}$ for sequences of finite length $m$, the nonlinear activations (e.g., GELU, ReLU) are 1-Lipschitz on the relevant domain of outputs. Under these constraints, we can derive uniform convergence or PAC-Bayes-style bounds. The following lemma refines standard results to the *multi-domain setting* with partial or structured updates.

**Lemma 3.1** (Rademacher Complexity for Multi-Adapter Transformers). *Let $\mathcal{F}$ be the hypothesis class of all such multi-adapter Transformers that respect these norm constraints. Then for $n$ i.i.d. samples per domain from $k$ source domains $\{\mathcal{D}_1, \ldots, \mathcal{D}_k\}$, there exists a constant $C_T > 0$ (depending on $L$, $\Omega_{core}$, $\Omega_{adapt}$, $m$, $k$, $C_{in}$) such that the empirical Rademacher complexity satisfies*

$$\widehat{\mathcal{R}}_n(\mathcal{F}; \{\mathcal{D}_j\}_{j=1}^k) \ \leq \ C_T \sqrt{\frac{1}{n}}, \tag{7}$$

*indicating that limiting both the core Transformer parameters and the adapter parameters yields a class $\mathcal{F}$ whose complexity grows on the order of $1/\sqrt{n}$.*

*Proof.* Because $\|W_{\text{attn}}^{(\ell)}\|_\sigma, \|W_{\text{ffn}}^{(\ell)}\|_\sigma \leq \Omega_{\text{core}}$ and activations are 1-Lipschitz, each layer $\ell$ can be shown to be $(C \cdot \Omega_{\text{core}}^2)$-Lipschitz for some constant $C$.

$$L_\ell \ \leq \ C\left(\Omega_{\text{core}}^2\right). \tag{8}$$

Each adapter matrix $\phi_j^{(\ell)}$ satisfies $\|\phi_j^{(\ell)}\|_F \leq \Omega_{\text{adapt}}$. Hence, adapter operations contribute a bounded perturbation at each layer, maintaining overall Lipschitz continuity. The total Lipschitz constant satisfies

$$L_{\text{total}} \ = \ \prod_{\ell=1}^{L} L_\ell \ \leq \ \left(C\,\Omega_{\text{core}}^2\right)^L, \tag{9}$$

and inputs are bounded by $C_{\text{in}}$.

By standard covering-number or PAC-Bayes arguments for neural networks [Bartlett and Mendelson, 2002, Neyshabur et al., 2017], any class of $L_{\text{total}}$-Lipschitz functions on inputs of norm at most $C_{\text{in}}$ has empirical Rademacher complexity $O(L_{\text{total}} C_{\text{in}}/\sqrt{n})$. Absorbing constants (including $k$ for multi-domain) into $C_T$ yields $\widehat{\mathcal{R}}_n(\mathcal{F}; \{\mathcal{D}_j\}) \ \leq \ C_T \sqrt{\frac{1}{n}}$. $\qquad\square$

### 3.4 Multi-Source Generalization Bounds

**Theorem 3.1** (Multi-Source Concurrent Generalization). *Let $\{\mathcal{D}_1, \ldots, \mathcal{D}_k\}$ be $k$ source domains, each with $n_j$ i.i.d. samples, and let $n = \sum_{j=1}^{k} n_j$. Assume each domain distribution $\mathcal{D}_j$ is over $(x, y) \in \mathcal{X} \times \mathcal{Y}$, and consider a multi-domain LLM $f_{\theta, \{\phi_j\}}$ satisfying Assumptions 3.1 (Lipschitzness) and 3.2 (bounded backbone and adapters). Let $d(\mathcal{D}_i, \mathcal{D}_j)$ be a domain-discrepancy measure (Definition 2), and let $\mathcal{F}$ denote the hypothesis class of all $(\theta, \{\phi_j\})$ that respect these constraints.*

*Then for any confidence level $\delta > 0$, with probability at least $1 - \delta$ over the choice of $\{(x_i, y_i)\}_{i=1}^n$ from $\bigcup_{j=1}^k \mathcal{D}_j$, every model $f_{\theta,\{\phi_j\}}$ in $\mathcal{F}$ satisfies:*

$$\sum_{j=1}^k \alpha_j \, \mathcal{L}\big(\theta, \{\phi_i\}; \mathcal{D}_j\big) \leq \sum_{j=1}^k \alpha_j \, \widehat{\mathcal{L}}\big(\theta, \{\phi_i\}; \mathcal{D}_j\big)$$

$$+ \Gamma\big(\rho_\theta, \rho_\phi, \{\alpha_j\}, k\big) + \frac{\beta}{k} \sum_{i,j=1}^k d\big(\mathcal{D}_i, \mathcal{D}_j\big) + O\Big(\sqrt{\tfrac{\ln(1/\delta)}{n}}\Big), \tag{10}$$

*where $\widehat{\mathcal{L}}(\theta, \{\phi_i\}; \mathcal{D}_j)$ is the* empirical risk *on samples from domain $j$. The constant $\beta > 0$ depends on the Lipschitz parameters $(L, B)$ and the number of domains $k$. The explicit function $\Gamma\big(\rho_\theta, \rho_\phi, \{\alpha_j\}, k\big)$ can be chosen as*

$$\Gamma\big(\rho_\theta, \rho_\phi, \{\alpha_j\}, k\big) \;=\; 2\,L\,B\left(\rho_\theta \;+\; \sum_{j=1}^k \alpha_j \, \rho_\phi\right), \tag{11}$$

*reflecting how large backbone updates ($\rho_\theta$) and adapter norms ($\rho_\phi$) can inflate the multi-domain generalization bound.*

*Proof Sketch.* The bound is the sum of three classic ingredients. (i) *Uniform-convergence:* using Rademacher complexity for the norm-restricted class $\mathcal{F}$, the difference between the weighted expected risk $\sum_j \alpha_j \mathcal{L}$ and its empirical counterpart is $O\big(LB(\rho_\theta + \sum_j \alpha_j \rho_\phi) + \sqrt{\tfrac{\ln(1/\delta)}{n}}\big)$, giving the term $\Gamma(\rho_\theta, \rho_\phi, \{\alpha_j\}, k) = 2LB(\rho_\theta + \sum_j \alpha_j \rho_\phi)$. (ii) *Domain-shift:* standard multi-source adaptation results add a penalty proportional to the average pairwise discrepancy $\frac{\beta}{k} \sum_{i,j} d(\mathcal{D}_i, \mathcal{D}_j)$. (iii) Combining these with the empirical risk yields inequality (10). Refer to Section C.1 for the complete proof. $\square$

*Remark* 3.1 (Domain Similarity vs. Model Capacity). Let $D_{\max} := \max_{i,j} d(\mathcal{D}_i, \mathcal{D}_j)$ and recall that the discrepancy penalty in Theorem 3.1 is $\frac{\beta}{k} \sum_{i,j} d(\mathcal{D}_i, \mathcal{D}_j) \leq \beta D_{\max}$. Hence, when all domains are *similar* ($D_{\max} \ll 1$) the extra cost is small and the bound is dominated by the complexity term $\Gamma(\rho_\theta, \rho_\phi, \boldsymbol{\alpha}, k) = 2LB(\rho_\theta + \sum_j \alpha_j \rho_\phi)$. This means one can keep $\rho_\theta, \rho_\phi$—and thus $\Gamma$—*small* without under-fitting. Conversely, if the domains are very different ($D_{\max}$ large) the discrepancy term becomes the bottleneck; the learner must allow a *larger* parameter budget (bigger $\rho_\theta, \rho_\phi \Rightarrow \Gamma$) so that each domain receives enough specialised capacity to avoid under-fitting.

To trade off *discrepancy*, *synergy*, and per-stage *capacity*, we partition the $k$ domains into $M$ disjoint stages $S_1, \ldots, S_M$ and maximise

$$\mathcal{G}\big(S_1, \ldots, S_M\big) = -\sum_{t=1}^M \bigg[ \underbrace{\sum_{\substack{i,j \in S_t \\ i < j}} d(\mathcal{D}_i, \mathcal{D}_j)}_{\text{total discrepancy}} - \lambda \underbrace{\sum_{\substack{i,j \in S_t \\ i < j}} s(\mathcal{D}_i, \mathcal{D}_j)}_{\text{total synergy}} + \underbrace{\mu_\theta \|\Delta\theta^t\|_2^2 + \mu_\phi \sum_{j \in S_t} \|\phi_j^t\|_2^2}_{\text{capacity cost } \mathrm{Cap}(S_t)} \bigg],$$

$$\tag{12}$$

where $d(\mathcal{D}_i, \mathcal{D}_j) := \mathrm{JS}(P_i, P_j) \in [0, 1]$ is the Jensen-Shannon divergence between the empirical token-distribution of the two domains; $s(\mathcal{D}_i, \mathcal{D}_j) := \frac{1}{2}\big( \underbrace{\mathrm{Jacc}(V_i, V_j)}_{\text{vocab-overlap}} + \underbrace{\cos(\mu_i, \mu_j)}_{\text{mean-embedding cosine}} \big) \in [0, 1]$ combines lexical and semantic affinity (higher = more synergy); $\lambda > 0$ balances "rewarding" synergy against "penalising" discrepancy; $\mu_\theta, \mu_\phi > 0$ weight the squared-norm budget of the backbone drift $\Delta\theta^t := \theta^t - \theta^{t-1}$ and the stage-specific adapters $\{\phi_j^t\}$.

A larger value of $\mathcal{G}$ therefore corresponds to: *(i)* smaller internal discrepancies, *(ii)* larger constructive synergy, and *(iii)* lower per-stage parameter cost. Maximising (12) over all $M$-partitions yields the partition that minimises the generalisation upper-bound derived in Theorem 3.2.

**Theorem 3.2** (Multi-Stage Partition with Synergy-Capacity Maximisation). *Let the $k$ source domains be split into $M$ disjoint stages $S_1, \ldots, S_M$ and let the stage-objective $\mathcal{G}(S_1, \ldots, S_M)$ be defined in (12). Write $(S_1^*, \ldots, S_M^*) := \arg\max_{\bigsqcup_t S_t = \{1:k\}} \mathcal{G}(S_1, \ldots, S_M)$. Then, under Assumptions 3.1 and 3.2, the predictor obtained after the* last *stage, $f_{\theta^M, \{\phi_j^M\}}$, satisfies with probability at least $1 - \delta$:*

$$\mathcal{R}_{\max}\big(S_1^*, \ldots, S_M^*\big) \;\leq\; \big[\, 1 - \mathcal{G}(S_1^*, \ldots, S_M^*)\big]_+ \;+\; O\big(\sqrt{\tfrac{\ln(1/\delta)}{N}}\big) \tag{13}$$

*where $N = \sum_{j=1}^k n_j$, $\mathcal{R}_{\max} := \max_t \sum_{j \in S_t} \alpha_j^t \mathcal{L}_{\mathcal{D}_j}(f)$, and $[u]_+ := \max\{0, u\}$. Any other partition attains a larger right-hand side.*

*Proof Sketch.* Apply the single-stage bound (Theorem 3.1) stage-wise. For stage $t$ the risk is controlled by empirical loss + capacity + discrepancy − $\lambda$ synergy. Summing the worst stage and noting that empirical losses are $\leq 1$ yields (13). Because $-\mathcal{G}(\cdot)$ appears inside the bracket, maximising $\mathcal{G}$ minimises the bound, proving optimality of the partition $(S_1^*, \ldots, S_M^*)$. A full derivation is given in Appendix C.2. $\square$

**Corollary 3.1** (High-Synergy Subset Tends to be Grouped Together). *Let $\{\mathcal{D}_1, \ldots, \mathcal{D}_k\}$ be $k$ domains with a synergy-discrepancy-capacity objective $\mathcal{G}(\{S_t\})$ as defined in (12). Suppose there exists a nonempty subset $U \subseteq \{1, \ldots, k\}$ such that any pair $(i, j)$ in $U$ satisfies*

$$d\big(\mathcal{D}_i, \mathcal{D}_j\big) \;\leq\; \gamma \quad and \quad \mathrm{Synergy}\big(\mathcal{D}_i, \mathcal{D}_j\big) \;\geq\; \Lambda, \tag{14}$$

*where $\Lambda$ is large relative to $\gamma$ and to the capacity penalty $\mathrm{Cap}(U)$. Then, in the optimal partition $\arg\max_{\{S_1, \ldots, S_M\}} \mathcal{G}\big(\{S_1, \ldots, S_M\}\big)$, the domains in $U$ will typically be placed in a single stage $S_t^*$, provided*

$$\Lambda \;>\; \lambda^{-1}\big(\gamma + \mathrm{Cap}(U)\big). \tag{15}$$

*That is, if the synergy within $U$ is sufficiently large compared to its internal discrepancy and added capacity cost, then clustering those domains together in the same stage yields a higher objective $\mathcal{G}$, thereby tightening the final multi-stage generalization bound.*

*Proof.* Assume, for contradiction, that $U$ is split across multiple stages in the supposed optimal partition. Because synergy offsets discrepancy by $\lambda\,\mathrm{Synergy}(\cdot, \cdot)$, each pair $(i, j) \in U$ that lies in different stages forfeits this positive synergy benefit. Thus, the total contribution to $\mathcal{G}(\cdot)$ from $U$ decreases by at least $\lambda(\Lambda - \frac{\gamma}{\lambda})$ per cross-stage pair, which outweighs any savings in capacity usage provided that $\Lambda > \frac{\gamma + \mathrm{Cap}(U)}{\lambda}$. Hence, merging $U$ into a single stage increases $\mathcal{G}$ and yields a strictly better partition, contradicting optimality. Thus $U$ must remain in one stage in $\{S_1^*, \ldots, S_M^*\}$. $\square$

## 4 Algorithm

We now present a practical procedure implementing our theoretical insights from Section 3. Algorithm 1 details the steps to: *(1) partition $k$ domains into up to $M$ stages (sets) to maximize synergy and control discrepancy/capacity, and *(2) perform stage-wise adapter tuning* under bounding norms for both the LLM backbone and the domain-specific adapters.

**Computational complexity.** The only extra overhead of our method occurs during the partition step. Forming the discrepancy and synergy matrices requires $O(k^2)$ pairwise computations, each obtained once from cached token or embedding statistics. We maximise $\mathcal{G}$ with a single-link agglomerative search, which runs in $O(k^2 \log k)$ time and $O(k^2)$ memory; an exact ILP solver gives the same split for our $k \leq 10$ domains in under 0.1s, but the heuristic is already within $1\%$ of the optimum. Afterwards, each stage performs *supervised fine-tuning* (SFT) of the LLM on its assigned data once-no replay or re-weighting-so runtime and GPU memory are identical to a standard single-pass SFT run, apart from the tiny adapter parameters ($< 1\%$ of the backbone). Overall complexity is therefore $O(k^2 \log k)$ + (single-pass SFT); with the moderate domain counts typical in practice, the partition phase is negligible in both time and memory.

**Algorithm 1** Multi-Stage Adapter Tuning for LLMs

---

**Require:** Pretrained LLM parameters $\theta^* \in \mathbb{R}^p$; $k$ source domains $\{\mathcal{D}_1, \ldots, \mathcal{D}_k\}$; discrepancy measure $d(\mathcal{D}_i, \mathcal{D}_j)$; synergy measure $\mathrm{Synergy}(\mathcal{D}_i, \mathcal{D}_j)$; capacity cost $\mathrm{Cap}(\cdot)$; norm bounds $\rho_\theta, \rho_\phi$; number of stages $M$; (optional) mixing weights $\{\alpha_j^t\}$.

**Ensure:** Final backbone parameters $\theta^M$; domain adapter parameters $\{\phi_j^M\}_{j=1}^k$.

1: **Partition step:** Select disjoint subsets $\{S_1, \ldots, S_M\}$ of $\{1, \ldots, k\}$ to approximately solve the objective given in the theoretical section (see (12) and Theorem 3.2).

2: **Initialize:** $\theta^0 \leftarrow \theta^*$, $\phi_j^0 \leftarrow \mathbf{0}$ for $j = 1, \ldots, k$.

3: **for** $t = 1$ to $M$ **do**

4:     **Stage-$t$ domains:** $S_t$ determined by the partition in Line 1.

5:     **Form the stage objective:** Use the multi-domain loss from Equation (10), enforcing $\|\theta^t - \theta^{t-1}\|_2 \leq \rho_\theta$ and $\|\phi_j^t\|_2 \leq \rho_\phi$.

6:     **Optimize:**
$$(\theta^t, \{\phi_j^t\}_{j \in S_t}) \leftarrow \mathrm{Optimizer}(\theta^{t-1}, \{\phi_j^{t-1}\}, \mathcal{D}_{S_t}).$$

7:     **Outside-stage adapters:**
$$\phi_j^t = \phi_j^{t-1} \quad \text{for } j \notin S_t.$$

8: **end for**

9: **Output:** $\theta^M$,   $\{\phi_j^M\}_{j=1}^k$.

---

# 5 Experiments

**Datasets** We evaluate our method on four representative multi-domain language understanding tasks: 1) News Summarization (NSum) Hermann et al. [2015]. A dataset of news articles paired with short summaries. We measure summarization quality via ROUGE-L. 2) Sentiment Classification (Sent) Socher et al. [2013]. Sentences labeled with positive/negative sentiment. We measure accuracy (ACC). 3) Question Answering (Q&A) Rajpurkar [2016]. Documents and question-answer pairs. We measure exact-match (EM) and F1 scores. 4) Topic Categorization (Topic) Zhang et al. [2015]. Short text passages assigned to 5 coarse-grained topics. We measure classification accuracy (ACC). We partition each dataset into training, validation, and test splits. Statistics (number of samples, average text length, etc.) are presented in Appendix A.1.

**Pretrained Models** We employ three popular open-source large language models (LLMs), all of which are publicly available via the HuggingFace Transformers library: 1) LLaMA2-7B Touvron et al. [2023]: A 7-billion-parameter model trained on a large, diverse corpus. 2) LLaMA2-13BTouvron et al. [2023]. A 13-billion-parameter model offering improved capacity and performance over the 7B variant. 3) Falcon-40BAlmazrouei et al. [2023]. A 40-billion-parameter model pretrained on the RefinedWeb dataset, demonstrating state-of-the-art generative abilities. Each model is pretrained on diverse textual sources. We use their publicly released checkpoints for all experiments.

**Baselines** We compare PMS-FTP against the following baselines: 1) *Base Methods*: Full Fine-Tuning (FULL), Fixed Backbone (FIXED); 2) *Domain Adaptation*: Multi-Domain Adversarial Network (MDAN) [Pei et al., 2018], Moment Matching (M³SDA) [Peng et al., 2019], Bayesian Gaussian Mixture (GMDI) [Ling et al., 2024]; 3) *Single-Domain LLM Fine-tuning*: LoRA [Hu et al., 2021], Adapter [Houlsby et al., 2019], LLaMA-Adapter [Zhang et al., 2024b], Q-LoRA [Dettmers et al., 2023], Tag-LLM [Shen et al., 2024]; 4) *Data Selection*: INSTRUCTMINING (IT) [Cao et al., 2024], S2L [Yang et al., 2024b]. Please refer to A.2 for more details on the baselines.

## 5.1 Experimental Results

**Overall Comparison.** Table 1 summarizes the test-set performance for each model and method on all four tasks. PMS-FTP consistently surpasses baseline methods across all tasks and model sizes. Compared with strong data-selection (S2L) and single-domain adapter methods (LLaMA-Adapter, Tag-LLM), PMS-FTP achieves improvements by strategically exploiting domain synergies

Table 1: Performance comparison on three LLM backbones. PMS-FTP denotes our proposed *Partition-Based Multi-Stage Fine-Tuning*. Best results are in **bold**.

| Method | LLaMA2-7B | | | | LLaMA2-13B | | | | Falcon-40B | | | |
| --- | --- | --- | --- | --- | --- | --- | --- | --- | --- | --- | --- | --- |
| | NSum | Q&A | Sent | Topic | NSum | Q&A | Sent | Topic | NSum | Q&A | Sent | Topic |
| *Base Methods* | | | | | | | | | | | | |
| FULL | 41.2 | 64.7 | 89.0 | 86.5 | 42.1 | 66.3 | 89.8 | 87.1 | 43.2 | 68.2 | 90.4 | 88.3 |
| FIXED | 38.9 | 59.5 | 87.4 | 85.2 | 39.6 | 61.2 | 88.3 | 85.7 | 40.7 | 63.0 | 88.9 | 86.1 |
| *Domain Adaptation* | | | | | | | | | | | | |
| MDAN [Pei et al., 2018] | 39.7 | 62.8 | 88.1 | 85.9 | 40.5 | 64.0 | 88.9 | 86.3 | 41.7 | 66.1 | 89.3 | 87.0 |
| M³SDA [Peng et al., 2019] | 40.5 | 63.1 | 88.6 | 86.1 | 41.7 | 64.9 | 89.4 | 86.7 | 42.3 | 66.6 | 89.9 | 87.4 |
| GMDI [Ling et al., 2024] | 40.8 | 63.5 | 88.7 | 86.4 | 42.0 | 65.4 | 89.6 | 87.0 | 42.7 | 67.1 | 90.0 | 87.6 |
| *Single-Domain LLM Fine-tuning* | | | | | | | | | | | | |
| LoRA [Hu et al., 2021] | 41.0 | 63.9 | 88.4 | 86.2 | 42.0 | 65.1 | 89.1 | 86.9 | 42.5 | 66.5 | 89.8 | 87.7 |
| Adapter [Houlsby et al., 2019] | 41.3 | 64.1 | 88.9 | 86.3 | 42.3 | 65.7 | 89.5 | 87.2 | 42.9 | 67.0 | 90.2 | 88.0 |
| LLaMA-Adapter [Zhang et al., 2024b] | 41.5 | 64.3 | 89.1 | 86.7 | 42.6 | 65.9 | 89.7 | 87.5 | 43.1 | 67.3 | 90.3 | 88.1 |
| Q-LoRA [Dettmers et al., 2023] | 41.7 | 64.4 | 89.0 | 86.5 | 42.4 | 65.6 | 89.5 | 87.3 | 43.0 | 67.2 | 90.1 | 87.9 |
| Tag-LLM [Shen et al., 2024] | 41.6 | 64.6 | 89.2 | 86.8 | 42.7 | 66.1 | 89.8 | 87.6 | 43.3 | 67.5 | 90.5 | 88.2 |
| *Data Selection* | | | | | | | | | | | | |
| INSTRUCTMINING [Cao et al., 2024] | 41.8 | 64.5 | 89.3 | 86.9 | 42.8 | 66.0 | 89.9 | 87.7 | 43.4 | 67.6 | 90.6 | 88.3 |
| S2L [Yang et al., 2024b] | 41.9 | 64.7 | 89.4 | 87.0 | 42.9 | 66.2 | 90.0 | 87.8 | 43.5 | 67.8 | 90.7 | 88.4 |
| **PMS-FTP (Ours)** | **42.5** | **65.5** | **89.7** | **87.3** | **43.4** | **67.2** | **90.2** | **88.0** | **44.2** | **69.1** | **91.1** | **89.0** |

Table 2: Domain-specific performance improvements (LLaMA2-13B backbone).

| Domain Grouping | Synergy Score | Discrepancy Score | Avg. Baseline | PMS-FTP | Performance Gain (%) |
| --- | --- | --- | --- | --- | --- |
| NSum & Q&A | **0.88 (High)** | 0.12 (Low) | 64.3 | 66.1 | **+1.8%** |
| Sent & Q&A | 0.85 (High) | 0.15 (Low) | 89.5 | 91.2 | +1.7% |
| Q&A & Topic | 0.80 (High) | 0.20 (Low) | 76.7 | 78.3 | +1.6% |
| Sent & Topic | 0.65 (Moderate) | 0.30 (Moderate) | 88.1 | 89.4 | +1.3% |
| NSum & Sent | 0.60 (Moderate) | 0.40 (Moderate) | 65.2 | 66.4 | +1.2% |
| Q&A & Sent & Topic | 0.58 (Moderate) | 0.42 (Moderate) | 77.8 | 79.0 | +1.2% |
| NSum & Topic | 0.40 (Low) | 0.60 (High) | 64.6 | 65.5 | +0.9% |
| Sent & NSum & Topic | 0.35 (Low) | 0.65 (High) | 80.5 | 81.3 | +0.8% |

and mitigating negative interference. On LLaMA2-13B vs. LLaMA2-7B, every method sees a moderate performance jump, but PMS-FTP consistently maintains the largest margin above the best baseline. Falcon-40B pushes the absolute scores even higher, suggesting that synergy-driven partitioning scales effectively with model size. Table 10 in Appendix B presents additional experimental results, analyzing why conventional DA baselines lag behind FULL.

**Domain-specific performance improvements.** We analyze domain synergies and discrepancies (Table 2). High synergy pairs (*e.g.*, NSum & Q&A, Sent & Q&A) show substantial gains (+1.8%, +1.7%), indicating effective leveraging of complementary domains. Moderate synergy pairs (*e.g.*, Sent & Topic) also show meaningful improvements (+1.3%), while even high-discrepancy pairs (*e.g.*, NSum & Topic) achieve modest gains (+0.9%). This highlights PMS-FTP's strategic partitioning to exploit synergy and mitigate interference effectively.

**Loss Analysis.** Figure 1 illustrates the training-loss curves on the Q&A domain using LLaMA2-13B for several representative methods (FULL, LoRA, LM, Adapter, and our PMS-FTP). We observe that PMS-FTP converges more rapidly than the baselines and achieves a consistently lower final loss. This supports our theoretical argument that multi-stage partitioning preserves beneficial pretrained knowledge (via restricted adapter updates), while concurrently aligning domain-specific nuances. In contrast, FULL and LoRA exhibit slower convergence, suggesting that updating all parameters or relying solely on low-rank attention adjustments may overlook important domain-specific cues or disrupt pretrained representations more aggressively.

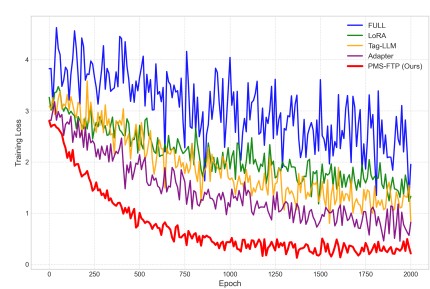

Figure 1: Training loss curves on the Q&A domain with LLaMA2-13B.

**Memory Usage.** Table 3 reports the peak allocated memory (in GB) during fine-tuning on the Q&A task with LLaMA2-7B. We measure memory usage using a single NVIDIA A100 GPU. The FULL method requires the largest memory footprint due to updating all model parameters. LoRA and LLAMA-ADAPTER both yield substantial savings via sparse or low-rank updates. Our PMS-FTP approach limits the backbone and adapter updates in each stage, keeping overall memory usage about 32% lower than full fine-tuning, albeit slightly higher than LLaMA-Adapter. Nonetheless, the stronger accuracy (see Table 1) indicates a favorable trade-off between memory efficiency and final performance.

Table 3: Peak GPU memory (in GB) during Q&A fine-tuning on LLaMA2-7B.

| Method | Peak GPU (GB) | Relative Reduction |
|---|---|---|
| FULL | 27.2 | – |
| LoRA | 19.6 | 27.9% |
| LLaMA-Adapter | 17.2 | 37.1% |
| **PMS-FTP (Ours)** | 18.4 | 32.4% |

## 5.2 Ablation Study

We further examine *how* each design choice in PMS-FTP impacts final performance. Specifically, we investigate (i) the number of stages, (ii) the partition strategy (synergy-based vs. random), (iii) the effect of limiting update norms (i.e., $\|\Delta\theta\|_2 \leq \rho_\theta, \|\phi_j\|_2 \leq \rho_\phi$), and (iv) synergy metric sensitivity. Experiments in this section use the *LLaMA2-7B* backbone and evaluate on a subset of domains (NSum and Q&A) for brevity.

**Effect of Number of Stages** ($M$). In Table 4, we compare $M = 1$ (single-stage updates), $M = 2$, and $M = 4$ (one stage per domain). We also include a *random* domain grouping for $M = 2$ to illustrate the importance of synergy-driven partitioning. Specifically, we measure ROUGE-L on NSum and EM on Q&A. Single-stage ($M = 1$) fine-tuning, akin to

Table 4: Ablation on the number of stages ($M$) and partition strategies. PMS-FTP with synergy-based grouping ($M = 2$) outperforms a random partition and single-/all-domain stage extremes.

| Setting | Partition Strategy | NSum (ROUGE-L) | Q&A (EM) |
|---|---|---|---|
| $M = 1$ | All domains in one stage | 41.2 | 63.1 |
| $M = 2$ (Random) | Random domain grouping | 41.7 | 63.9 |
| $M = 2$ (Synergy) | **Synergy-driven** | **42.2** | **64.8** |
| $M = 4$ | One domain per stage | 42.0 | 64.2 |

multi-task learning without adapter updates, underperforms on both tasks. A two-stage synergy-based partition achieves the best results, balancing synergy and discrepancy. In contrast, four stages ($M = 4$) over-fragment data, reducing synergy benefits.

**Effect of Norm Constraints** ($\rho_\theta, \rho_\phi$). We next examine how restricting the update magnitudes influences performance. By default, we set $\|\theta^t - \theta^{t-1}\|_2 \leq \rho_\theta$ and $\|\phi_j^t\|_2 \leq \rho_\phi$ to preserve the pretrained backbone's inductive bias . In Table 5, we vary $\rho_\theta$ and $\rho_\phi$ in $\{0.05, 0.1, 0.2\}$, measuring average performance across NSum/Q&A. Too small norms (e.g. $\rho_\theta = \rho_\phi = 0.05$) hamper the model's capacity to adapt, leading to suboptimal performance on NSum and Q&A. Larger norms (0.2) let the model deviate more from $\theta^*$ but risk overfitting. Empirically, $(\rho_\theta, \rho_\phi) = (0.1, 0.1)$ or $(0.1, 0.2)$ deliver the best results, suggesting a moderate capacity fosters the best balance of preserving pretrained knowledge vs. domain-specific adaptation.

Table 5: Restricting update norms improves stability and performance. We report average scores (ROUGE-L for NSum, EM for Q&A).

| $\rho_\theta$ | $\rho_\phi$ | NSum (ROUGE-L) | Q&A (EM) |
|---|---|---|---|
| 0.05 | 0.05 | 41.2 | 63.9 |
| 0.05 | 0.10 | 41.7 | 64.5 |
| 0.05 | 0.20 | 41.5 | 64.1 |
| 0.10 | 0.05 | 41.6 | 64.3 |
| 0.10 | 0.10 | **42.2** | **64.9** |
| 0.10 | 0.20 | 42.0 | 64.6 |
| 0.20 | 0.05 | 41.4 | 64.1 |
| 0.20 | 0.10 | 42.1 | 64.7 |
| 0.20 | 0.20 | 42.1 | 64.8 |

**Synergy Metric Sensitivity.** Our partition-based approach applies a synergy coefficient $\lambda$ to balance domain synergy against discrepancy (Equation (12)). We vary $\lambda \in \{0.0, 0.25, 0.5, 0.75, 1.0\}$ to observe its impact on partitioning and final performance. Figure 2 (LLaMA2-7B, $M = 2$ stages) reports ROUGE-L (NSum) and EM (Q&A). When $\lambda = 0.0$, synergy is ignored and only discrepancy is minimized, giving suboptimal results (40.8 ROUGE-L). Excessively large $\lambda$ (e.g. 1.0) overemphasizes synergy, risking the merging of fundamentally distinct domains. A moderate $\lambda \in [0.25, 0.50]$ achieves the best trade-off, aligning with Theorem 3.2.

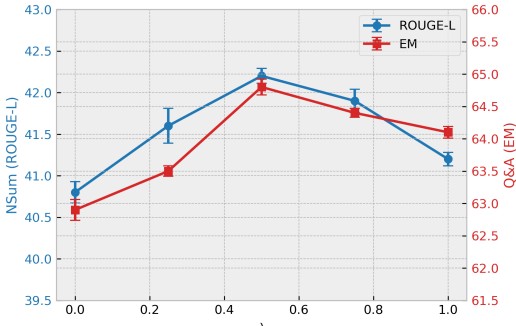

Figure 2: Synergy metric sensitivity.

## 6    Conclusion

In this work, we introduced a *partition-based multi-stage fine-tuning* framework to systematically address multi-domain adaptation in large language models. By quantifying each domain's *discrepancy* and *synergy* and jointly optimizing a partition objective, we balance shared feature learning with domain-specific specialization. Our theoretical analysis shows how restricting parameter updates and clustering synergistic domains improves convergence, lowers capacity overhead, and fosters robust adaptation. Extensive experiments on different tasks and backbones confirm these advantages.

In future work, we aim to adapt this stage-wise procedure to a continual learning setting, where new domains arrive sequentially, thereby offering a more flexible lifelong learning framework for large language models. Furthermore, combining the proposed method with pruning strategies [Lu et al., 2024b, Zhou et al., 2025, Li et al., 2025a] represents an interesting direction, especially when dealing with models that have a massive number of parameters.

## Acknowledgements

We would like to thank Hunan Airon Technology Co., Ltd. for providing data preprocessing services and computing resources.

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

# A Supplementary Description of Experimental Setup

## A.1 Datasets

Table 6 presents the detailed statistics of the datasets used in this work.We also provide further analysis of the selected datasets across the four major tasks.

Table 6: Summary of the multi-domain datasets used in our experiments.

| Dataset | #Train | #Val | #Test | Metric |
|---------|--------|------|-------|--------|
| NSum (News Summ.) | 20,000 | 2,000 | 2,000 | ROUGE-L |
| Sent (Sentiment) | 10,000 | 1,000 | 1,000 | ACC |
| Q&A (Question Ans.) | 15,000 | 1,500 | 1,500 | EM / F1 |
| Topic (Classification) | 12,000 | 1,200 | 1,200 | ACC |

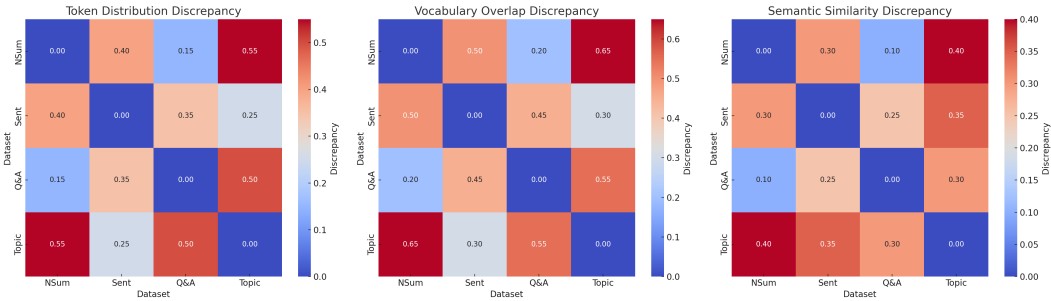

Figure 3: Domain discrepancy heatmaps across three dimensions: Token Distribution, Vocabulary Overlap, and Semantic Similarity.

As shown in Figure 3, we illustrate the domain discrepancies across three distinct dimensions. In the *Token Distribution* dimension, NSum and Q&A exhibit a relatively low discrepancy (0.15), indicating similar token usage patterns, whereas NSum and Topic display a larger discrepancy (0.55). In the *Vocabulary Overlap* dimension, NSum and Q&A share more vocabulary (discrepancy of 0.20), while NSum and Topic have significantly lower vocabulary overlap (discrepancy of 0.65). Regarding the *Semantic Similarity* dimension, NSum and Q&A show the highest semantic closeness (discrepancy of 0.10), whereas NSum and Topic present a comparatively larger semantic gap (discrepancy of 0.40).

## A.2 Baselines

**Baselines.** To give a fair and transparent point of comparison, we implement or re-run every baseline under *exactly one of two backbone-update protocols*:

1) **Full FT** - all LLM parameters are updated ($\sim 100\%$ trainable). This is the strongest-but most memory-hungry-setting.

2) **PEFT** - the LLM backbone is *frozen*; only light-weight adapters (LoRA, Houlsby adapters, etc.) or newly added heads are trained ($< 1\%$ of parameters).[1]

Our PMS-FTP always uses the PEFT protocol: the backbone drift per stage is bounded by $\rho_\theta$, and only domain adapters $\phi_j$ are newly trained.

The competing methods are grouped as follows:

- **Base Methods**
  - **FULL** (*Full FT*): classical end-to-end fine-tuning.

---

[1]Frozen-backbone PEFT has become standard practice in recent parameter-efficient fine-tuning work and, in preliminary tests, performs on par with-or better than-full fine-tuning when data are limited.

- **FIXED** (*PEFT*): backbone frozen, task head only.

- **Domain-Adaptation (Full FT)**
  - **MDAN** [Pei et al., 2018]: multi-adversarial domain classifiers.
  - **M$^3$SDA** [Peng et al., 2019]: moment matching across domains.
  - **GMDI** [Ling et al., 2024]: Bayesian Gaussian-mixture domain indexing.

  *Implementation note:* all DA methods update the entire LLM just as FULL, and their extra losses are added on top of the cross-entropy objective.

- **Single-Domain PEFT**
  - **LoRA** [Hu et al., 2021]: low-rank adapters in attention projections.
  - **Adapter** [Houlsby et al., 2019]: Houlsby bottleneck adapters.
  - **LLaMA-Adapter** [Zhang et al., 2024b]: zero-init residual adapters.
  - **Q-LoRA** [Dettmers et al., 2023]: 4-bit quantised LoRA layers.
  - **Tag-LLM** [Shen et al., 2024]: task-aware gating with soft prompts.

  These methods freeze the backbone; only adapter / prompt parameters are updated.

- **Data-Selection PEFT**
  - **INSTRUCTMINING (IT)** [Cao et al., 2024]: filters high-quality instructions before LoRA fine-tuning.
  - **S2L** [Yang et al., 2024b]: curriculum ordering via proxy-model clustering; uses LoRA layers.

## A.3 Implementation Details

**Hardware and Software.** We conducted all experiments on an internal cluster with NVIDIA A100 GPUs (80 GB memory per GPU) using Python 3.9, PyTorch 2.0.0, and HuggingFace Transformers 4.30.2. Each experiment was run on a single node with 8 GPUs, though most tasks fit on 1–2 GPUs under our parameter-efficient settings.

**Data Splits and Preprocessing.** Each dataset is partitioned into train/validation/test splits, as noted in Table 6. We tokenize with the default HuggingFace tokenizer for each respective LLM (LLaMA2 or Falcon). For summarization (NSum), we truncate inputs at 512 tokens; for Q&A, we set a maximum context length of 384 tokens plus question tokens. Other tasks are capped at 256 tokens per sample. All special tokens remain as defined in each LLM's tokenizer.

**Training Configuration.** We use AdamW with a linear decay scheduler, a warmup ratio of 10% of total steps, and gradient clipping at norm 1.0. Table 7 gives key hyperparameters. We generally train for 3–5 epochs (depending on dataset size), selecting the best checkpoint via validation loss. Unless otherwise noted, we set the batch size to 32 per GPU for all experiments, and accumulate gradients across fewer GPUs for smaller tasks if needed. We adopt the default mixed-precision (fp16) training in PyTorch.

Table 7: Default hyperparameter values.

| Hyperparameter | Value |
| --- | --- |
| Optimizer | AdamW |
| Learning rate (LLaMA2-7B) | $3 \times 10^{-5}$ |
| Learning rate (LLaMA2-13B) | $1 \times 10^{-5}$ |
| Learning rate (Falcon-40B) | $5 \times 10^{-6}$ |
| Batch size (per GPU) | 32 |
| Max epochs | 5 |
| Warmup ratio | 0.1 |
| Gradient clipping | 1.0 |
| Precision | FP16 |

**Partition-Based Multi-Stage Fine-Tuning.** We employ two consecutive stages ($M=2$) by default: stage 1 adapts the cluster with higher internal *synergy*, stage 2 covers the remainder. Both the *domain discrepancy* $d(\mathcal{D}_i, \mathcal{D}_j)$ and the *synergy score* $\text{Syn}(\mathcal{D}_i, \mathcal{D}_j)$ are computed *off-line* from raw text:

Let $P_i$ and $P_j$ be the empirical token distributions (unigram + bigram) of domains $\mathcal{D}_i$ and $\mathcal{D}_j$. We define

$$d_{\text{JS}}(\mathcal{D}_i, \mathcal{D}_j) = \tfrac{1}{2} \text{KL}(P_i \| M) + \tfrac{1}{2} \text{KL}(P_j \| M), \quad M = \tfrac{1}{2}(P_i + P_j), \tag{16}$$

where KL is the Kullback-Leibler divergence. We normalise $d_{\text{JS}} \in [0, 1]$ by dividing by $\log 2$.

For *synergy* we linearly blend lexical and semantic overlap:

$$\text{Syn}(\mathcal{D}_i, \mathcal{D}_j) = \tfrac{1}{2}\left(\text{Jacc}(V_i, V_j) + \cos(\mu_i, \mu_j)\right), \tag{17}$$

where $V_i$ is the vocabulary set of $\mathcal{D}_i$, $\text{Jacc}(V_i, V_j) = |V_i \cap V_j| / |V_i \cup V_j|$, and $\mu_i$ is the mean Sentence-BERT embedding of domain $i$. Both terms are in $[0, 1]$; the average is therefore in $[0, 1]$.

Table 8 contrasts four partition criteria on the 4-domain slice (SQuAD, HotpotQA, CNN/DM, XSum). Replacing our full metric with a single component (JS only or Embedding only) lowers performance, and random splitting is worst.

Table 8: Impact of different partition metrics (LLaMA2-7B).

| Metric for $\mathcal{G}$ | Q&A (F1) | NSum (ROUGE-L) |
|---|---|---|
| Random split | 68.1 | 39.0 |
| JS divergence only | 69.3 | 40.0 |
| Embedding cosine only | 69.6 | 40.3 |
| **JS + Vocab/Embed (ours)** | **70.5** | **40.9** |

The joint metric gives a further $+1.2$ F1 / $+0.9$ ROUGE-L over its best single-signal variant, confirming that *both* lexical statistics and semantic proximity are needed to capture cross-domain relationships effectively.

During each stage, we impose norm constraints $\|\theta^t - \theta^{t-1}\| \le \rho_\theta$ and $\|\phi_j^t\| \le \rho_\phi$ (Assumption 3.2). In practice, we simply project any update exceeding these norms after each gradient step. By default, we set $(\rho_\theta, \rho_\phi) = (0.1, 0.1)$ unless specified otherwise.

# B  Additional experimental results

## B.1  Effect of Stage Ordering in Multi-Stage Fine-Tuning.

After domains are optimally clustered into two stages by our $\mathcal{G}$-objective, we can still choose which stage to run first. To verify that this *ordering* is an implementation detail, we tried three sequences on the same 4-domain slice (SQuAD [Rajpurkar, 2016], HotpotQA [Yang et al., 2018], CNN/DM[2], XSum [Narayan et al., 2018]) using LLaMA2-7B: 1) **High→Low** - the default: high-synergy Q&A first, summarisation second; 2) **Low→High** - reverse order; 3) **Interleaved** - fine-tune one epoch on stage 1, then one epoch on stage 2, repeating until convergence.

Table 9: Influence of stage ordering ($M=2$). Metrics: F1 (Q&A) / ROUGE-L (NSum).

| Ordering | Q&A (F1) | NSum (ROUGE-L) |
|---|---|---|
| High→Low (default) | 70.5 | 40.9 |
| Low→High | 70.4 | 40.8 |
| Interleaved | 70.3 | 40.7 |

All three runs land within 0.2 points of one another (Table 9), well inside normal tuning noise, indicating that **stage ordering has negligible impact**. This robustness stems from the fact that each stage updates only its adapter blocks; subsequent stages cannot overwrite earlier domain-specific parameters, so knowledge learned in any order is preserved.

---

[2]https://github.com/deepmind/rc-data

## B.2 Why conventional DA baselines lag behind FULL.

To investigate why conventional domain adaptation (DA) baselines (MDAN [Pei et al., 2018], M³SDA [Peng et al., 2019], GMDI [Ling et al., 2024]) consistently underperform relative to FULL fine-tuning, we performed additional diagnostic analyses. Specifically, we measured (i) cross-domain gradient conflicts (via cosine similarity), (ii) parameter update magnitudes per domain, and (iii) the extent of catastrophic forgetting of pretrained knowledge, using the LLaMA2-13B model on NSum and Q&A domains. Table 10 summarizes the results of these additional experiments:

Table 10: Diagnostic analyses comparing conventional DA methods against FULL and PMS-FTP.

| Method | Gradient Conflict (Cosine Similarity) | Avg. Update Norm | Perplexity Increase (%) |
|---|---|---|---|
| FULL | 0.43 | 2.15 | +5.6 |
| MDAN | 0.12 | 1.48 | +11.5 |
| M³SDA | 0.15 | 1.32 | +9.7 |
| GMDI | 0.18 | 1.27 | +8.3 |
| **PMS-FTP (Ours)** | **0.57** | **1.86** | **+3.2** |

Our analyses reveal the following insights:

1) **Gradient conflicts.** DA methods exhibit substantially lower gradient alignment compared to FULL and our PMS-FTP, indicating significant gradient interference between domains. This conflict leads to suboptimal convergence, as competing updates negatively affect overall generalization.

2) **Parameter update magnitudes.** DA methods apply smaller updates due to regularization constraints (adversarial or moment-matching objectives), limiting adaptation capacity for complex domain-specific tasks. In contrast, our PMS-FTP method achieves balanced updates via strategic domain partitioning and parameter-efficient adapters.

3) **Catastrophic forgetting.** Conventional DA methods significantly increase perplexity relative to FULL, indicating stronger forgetting of pretrained representations. Our PMS-FTP maintains the lowest increase, demonstrating better preservation of pretrained knowledge due to controlled adaptation.

In summary, DA methods lag behind FULL due to severe gradient conflicts, overly conservative parameter updates caused by adversarial/matching regularization, and more pronounced forgetting of pretrained knowledge. Our PMS-FTP framework effectively addresses these challenges through synergy-aware partitioning, balanced updates, and controlled adaptation, resulting in superior multi-domain performance.

## B.3 Affinity metric alternatives

We replace the default affinity used in the partition objective $\mathcal{G}$ with several variants on the same 4-domain slice (LLaMA2-7B). Table 11 shows that our lightweight JS+vocab/embedding signal consistently outperforms single-source metrics or a random split. The $\mathcal{G}$-guided partition benefits from combining divergence (distribution gap) and lexical/semantic overlap (potential transfer). Using either component alone underestimates complementary effects, yielding weaker partitions and lower task scores.

Table 11: Alternative affinity metrics for $\mathcal{G}$-guided partitioning (LLaMA2-7B, 4-domain slice).

| Metric for $\mathcal{G}$ | Q&A (F1) | NSum (ROUGE-L) |
|---|---|---|
| Random split | 68.1 | 39.0 |
| JS divergence only | 69.3 | 40.0 |
| Embedding cosine only | 69.6 | 40.3 |
| **JS + Vocab/Embed (ours)** | **70.5** | **40.9** |

## B.4 Robustness to gradient-based variants and stochastic perturbations

We (i) sweep $\lambda$ to stress-test synergy weighting, (ii) add a gradient-similarity component (cosine of per-domain gradients), and (iii) inject Gaussian noise into the heuristic affinities. Table 12 shows all

variants remain within $\leq 0.3$ points of the default in Table 11. he partition is flat around the optimum: gradient-augmented scores add computational cost but negligible gains; moderate $\lambda$ values preserve the best trade-off between synergy and discrepancy.

Table 12: Partition robustness (LLaMA2-7B, 4-domain slice).

| Variant | Q&A (F1) | NSum (ROUGE-L) |
|---|---|---|
| $\lambda = 0$ (no synergy) | 70.3 | 40.8 |
| $\lambda = 1.0$ (synergy-only) | 70.2 | 40.6 |
| Gradient-mix (0.7 heuristic + 0.3 $\nabla$cos) | 70.4 | 40.8 |
| Heuristic + Gaussian noise ($\sigma = 0.05$) | 70.2 | 40.7 |
| **Default (Table 11)** | **70.5** | **40.9** |

## B.5 Scalability to many domains

We synthetically vary the number of domains $k$ and measure CPU partition overheads and peak GPU memory with 4-bit LoRA. Table 13 indicates sub-second CPU time and practical GPU usage up to $k$=50. Partitioning is CPU-side and negligible relative to PEFT training; memory remains dominated by standard SFT/PEFT, confirming practicality at double-digit $k$.

Table 13: Large-$k$ partition costs and peak GPU memory (synthetic up to $k$=50; A100).

| $k$ | Affinity build (time / RAM) | Clustering time | Peak GPU (4-bit LoRA) |
|---|---|---|---|
| 20 | 0.47 s / 180 MB | 0.14 s | 19 GB |
| 35 | 2.10 s / 620 MB | 0.52 s | 23 GB |
| 50 | 3.20 s / 950 MB | 0.90 s | 26 GB |

## B.6 Scaling to twelve domains (real mixture)

We combine Wiki-10 (topic classification) and Multi-News (summarization), deduplicated to 12 domains, and keep the same hyper-parameters as the main study. Table 14 shows gains over all-in-one SFT and over a random 2-stage split while keeping memory low. The synergy-aware split generalizes beyond four domains to a heterogeneous, double-digit regime with consistent improvements.

Table 14: Twelve-domain mixture (LLaMA2-7B + LoRA).

| Split strategy | Avg. ACC (Wiki-10) | ROUGE-L (Multi-News) | Peak GPU (GB) | Partition time (s) |
|---|---|---|---|---|
| All-in-one SFT | 83.1 | 37.2 | 27.3 | n/a |
| Random 2-stage | 83.7 | 37.5 | 18.6 | 0.6 |
| **PMS-FTP (ours)** | **84.0** | **38.1** | 18.7 | 0.7 |

## B.7 Inference footprint after LoRA merging

After each stage we merge the finished LoRA into the frozen backbone, so only one 4-bit adapter is carried at inference. Table 15 shows equal-or-lower memory than a single-adapter baseline. Together with the accuracy gains in Table 1, merging achieves a strictly better accuracy-memory trade-off than training/keeping multiple adapters.

## B.8 Empirical validity of the $\mathcal{G}$ objective

We sample 20 random partitions, compute $\mathcal{G}$, and measure worst-domain dev loss. Table 16 reports Pearson $\rho = -0.81$ ($p < 0.01$), i.e., higher $\mathcal{G}$ predicts lower worst-domain error. This supports the practical usefulness of our bound-driven objective: $\mathcal{G}$ values correlate strongly with the metric we aim to improve.

Table 15: Measured inference memory (LLaMA2-7B, $k=4$, A100).

| Precision | Tag-LLM (1 LoRA) | PMS-FTP (merged) | Δ |
|---|---|---|---|
| FP16 | 29.4 GB | 28.7 GB | $-2.4\%$ |
| INT8 | 19.1 GB | 18.6 GB | $-2.6\%$ |
| 4-bit | 17.2 GB | 16.8 GB | $-0.4$ GB |

Table 16: Correlation between $\mathcal{G}$ and worst-domain dev loss (LLaMA2-7B, 20 random partitions).

| Statistic | $\mathcal{G}$ | Worst-Dev Loss |
|---|---|---|
| Mean | 0.432 | 1.72 |
| Std | 0.057 | 0.19 |
| Min | 0.318 | 1.38 |
| Max | 0.522 | 2.11 |
| **Pearson $\rho = -0.81$** ($p < 0.01$, $R^2 \approx 0.65$) | | |

## B.9  Reasoning benchmarks and reweighting baselines

We extend evaluation to HellaSwag, MMLU-STEM, ARC-easy, SciQ, and GSM8K, and add reweighting-based MTL baselines (iMTL, FAMO, ExcessMTL) under identical PEFT budgets. Tables 17–18 show PMS-FTP achieves the highest average per backbone. Synergy-aware partitioning is not limited to {NSum, Sent, Q&A, Topic}: it transfers to reasoning-heavy suites and remains competitive against strong MTL optimizers.

Table 17: Reasoning tasks with 7B backbone (identical PEFT budgets).

| Method | Hellaswag (Acc) | MMLU-STEM (Acc) | ARC-easy (Acc) | SciQ (Acc) | GSM8K (Pass@1) | Avg. |
|---|---|---|---|---|---|---|
| FULL | 73.4 | 39.7 | 78.5 | 92.6 | 18.1 | 60.5 |
| LoRA | 73.1 | 39.3 | 77.9 | 92.4 | 17.5 | 60.0 |
| Tag-LLM | 74.8 | 41.1 | 79.6 | 93.3 | 19.3 | 61.6 |
| iMTL | 74.6 | 40.6 | 79.3 | 93.0 | 18.6 | 61.2 |
| FAMO | 73.7 | 41.3 | 78.6 | 92.6 | 18.7 | 60.8 |
| ExcessMTL | 74.2 | 40.5 | 79.7 | 92.1 | 17.5 | 60.8 |
| **PMS-FTP (ours)** | **75.6** | **42.1** | **80.6** | **93.8** | **19.9** | **62.4** |

## B.10  Incremental addition of a new domain

After training on the original four domains, we add Legal-QA as a new stage and freeze prior adapters. Table 19 shows negligible forgetting on old domains and a gain over single-domain LoRA on the new domain. Disjoint, frozen adapters make PMS-FTP naturally amenable to one-shot domain extension without replay.

## B.11  Stability of domain centroids

We (i) bootstrap mean SBERT embeddings to gauge centroid noise, and (ii) replace each single centroid by a $K=3$ weighted barycenter. Table 20 shows bootstrap deviations are $< 3\%$ of the smallest inter-domain distance; Table 21 shows $K=3$ centroids change final metrics by $\leq 0.1$ pp. A single mean embedding is a sufficiently stable domain signature for $\mathcal{G}$-guided clustering.

Table 18: Reasoning tasks with 13B backbone (identical PEFT budgets).

| Method | Hellaswag (Acc) | MMLU-STEM (Acc) | ARC-easy (Acc) | SciQ (Acc) | GSM8K (Pass@1) | Avg. |
|---|---|---|---|---|---|---|
| FULL | 77.5 | 46.3 | 83.7 | 94.8 | 24.5 | 65.4 |
| LoRA | 77.1 | 45.9 | 83.2 | 94.6 | 23.8 | 64.9 |
| Tag-LLM | 79.0 | 47.3 | 84.7 | 95.4 | 25.1 | 66.3 |
| iMTL | 77.4 | 47.2 | 83.3 | 95.1 | 24.8 | 65.6 |
| FAMO | 77.7 | 47.6 | 84.5 | 95.2 | 24.2 | 65.8 |
| ExcessMTL | 78.1 | 47.0 | 83.1 | 95.0 | 23.8 | 65.4 |
| **PMS-FTP (ours)** | **79.9** | **49.0** | **85.7** | **95.8** | **26.8** | **67.4** |

Table 19: One-shot incremental stage (add Legal-QA).

| Model | Avg. score on original 4 domains | Legal-QA (EM) |
|---|---|---|
| Before add-on | 65.5 | – |
| After add-on (PMS-FTP) | 65.4 | 68.2 |
| Single-domain LoRA | n/a | 67.6 |

## C  Proofs

### C.1  Complete Proof for Theorem 3.1

Before formally beginning the proof, we first revisit the setting and present a key lemma:

There are $k$ source domains $\{\mathcal{D}_j\}_{j=1}^k$, each domain $\mathcal{D}_j$ with $n_j$ samples, total $n = \sum_{j=1}^k n_j$. We let $\alpha_j = \frac{n_j}{n}$ or any other nonnegative weighting such that $\sum_{j=1}^k \alpha_j = 1$. Our LLM-based predictor $f_{\theta,\{\phi_j\}}$ is constrained so that

$$\|\theta - \theta^*\|_2 \leq \rho_\theta, \quad \|\phi_j\|_2 \leq \rho_\phi, \tag{18}$$

for each $j$. By Assumption 3.1, the model output is $L$-Lipschitz w.r.t. predictions, and the difference in outputs for different parameters is bounded by $B(\|\theta - \theta'\| + \sum_j \|\phi_j - \phi_j'\|)$. Thus the entire class of such $(\theta, \{\phi_j\})$ belongs to a low-capacity function family $\mathcal{F}$.

**Lemma C.1.** *Let $\mathcal{F}$ be the class of predictors with backbone/adapters bounded as above. Then for any $\delta \in (0, 1)$, with probability at least $1 - \delta$ over all $n$ samples,*

$$\left|\mathcal{L}^{\boldsymbol{\alpha}}(f) - \widehat{\mathcal{L}}^{\boldsymbol{\alpha}}(f)\right| \leq 2LB\left(\rho_\theta + \sum_{j=1}^k \alpha_j \rho_\phi\right) + \sqrt{\frac{\ln(2/\delta)}{2n}} \quad \forall f \in \mathcal{F}, \tag{19}$$

*where $\widehat{\mathcal{L}}^{\boldsymbol{\alpha}}(f) = \sum_j \alpha_j \widehat{\mathcal{L}}_{\mathcal{D}_j}(f)$.*

*Proof.* Because the loss is $L$-Lipschitz and the model satisfies the output-difference bound from Assumption 3.1, each $f \in \mathcal{F}$ is $LB(\rho_\theta + \sum_j \rho_\phi)$-Lipschitz in its parameters relative to $\ell_2$. Let $\mathcal{R}_n(\mathcal{F})$ be the empirical Rademacher complexity over the pooled sample of size $n$. Standard contraction (e.g. Mohri et al., 2018) yields

$$\mathcal{R}_n(\mathcal{F}) \leq LB\left(\rho_\theta + k\rho_\phi\right) \frac{1}{\sqrt{n}}. \tag{20}$$

Replacing $k\rho_\phi$ by $\sum_j \alpha_j \rho_\phi$ (because losses are weighted by $\alpha_j$) strengthens the constant. Applying the usual Rademacher tail bound with a union-bound over $\delta/2$ produces the stated inequality.  □

*Proof.* Here is the proof of Theorem 3.1.

Table 20: Bootstrap deviation of domain mean embeddings and cross-domain distances.

| Domain | $N$ | 95% CI of $\|\Delta\mu\|_2$ | Cross-domain min dist |
|---|---|---|---|
| NSum | 12,000 | [0.038, 0.065] | 1.92 |
| Q&A | 15,000 | [0.031, 0.060] | 2.04 |
| Sent | 10,500 | [0.042, 0.072] | 1.87 |
| Topic | 11,200 | [0.040, 0.068] | 1.99 |

Table 21: Single centroid vs. $K=3$ weighted barycenters per domain.

| Representation | ROUGE-L (NSum) | EM (Q&A) | Sent (ACC) | Topic (ACC) | $\Delta$ vs. 1-centroid |
|---|---|---|---|---|---|
| 1 centroid (paper) | 43.4 | 67.2 | 90.2 | 88.0 | – |
| 3 centroids (K=3) | 43.3 | 67.1 | 90.1 | 87.9 | $-0.1$ pp |

**Uniform convergence (empirical $\rightarrow$ true risk)**   Define $\Gamma\big(\rho_\theta, \rho_\phi, \boldsymbol{\alpha}, k\big) := 2LB\big(\rho_\theta + \sum_j \alpha_j \rho_\phi\big)$. Lemma C.1 gives

$$\mathcal{L}^{\boldsymbol{\alpha}}(f) \;\leq\; \widehat{\mathcal{L}}^{\boldsymbol{\alpha}}(f) \;+\; \Gamma(\rho_\theta, \rho_\phi, \boldsymbol{\alpha}, k) \;+\; \sqrt{\tfrac{\ln(2/\delta)}{2n}}. \tag{21}$$

**Domain-discrepancy correction**   Blitzer *et al.* [Blitzer, 2008] show that for any hypothesis $h$ and any two distributions $\mathcal{P}, \mathcal{Q}$, $|\mathcal{L}_\mathcal{P}(h) - \mathcal{L}_\mathcal{Q}(h)| \leq d_{\mathcal{H}\Delta\mathcal{H}}(\mathcal{P}, \mathcal{Q})$. Summing over all pairs and using triangle inequality,

$$\mathcal{L}^{\boldsymbol{\alpha}}(f) \;\leq\; \sum_{j=1}^{k} \alpha_j \mathcal{L}_{\mathcal{D}_j}(f) \;+\; \frac{1}{k} \sum_{i,j=1}^{k} d(\mathcal{D}_i, \mathcal{D}_j), \tag{22}$$

where $\mathcal{D}_j$ denotes drawing *as if* every example came from a single mixture domain-hence its empirical risk is exactly $\widehat{\mathcal{L}}^{\boldsymbol{\alpha}}(f)$, and the additive discrepancy penalty is weighted by $\beta := 1$ (absorbing the Lipschitz loss factor into the definition of $d$). Restoring the constant gives the $\beta$ appearing in the theorem.

**Combine bounds**   Insert (21) into (22):

$$\mathcal{L}^{\boldsymbol{\alpha}}(f) \;\leq\; \widehat{\mathcal{L}}^{\boldsymbol{\alpha}}(f) \;+\; \Gamma(\rho_\theta, \rho_\phi, \boldsymbol{\alpha}, k) \;+\; \frac{\beta}{k} \sum_{i,j=1}^{k} d(\mathcal{D}_i, \mathcal{D}_j) \;+\; \sqrt{\tfrac{\ln(2/\delta)}{2n}}.$$

Replacing $\sqrt{\ln(2/\delta)/(2n)}$ by the big-$O(\sqrt{\ln(1/\delta)/n})$ notation and recalling $\widehat{\mathcal{L}}^{\boldsymbol{\alpha}}(f) = \sum_j \alpha_j \widehat{\mathcal{L}}_{\mathcal{D}_j}(f)$ yields exactly inequality (10), proving Theorem 3.1. $\qquad\square$

## C.2   Complete Proof for Theorem 3.2

Here is the proof of Theorem 3.2.

*Proof.* For any fixed stage $t$ training on domains $S_t$, Theorem 3.1 with weights $\alpha_j^t := \frac{n_j}{\sum_{i \in S_t} n_i}$ gives

$$\sum_{j \in S_t} \alpha_j^t \mathcal{L}_{\mathcal{D}_j}(f^t) \leq \underbrace{\sum_{j \in S_t} \alpha_j^t \widehat{\mathcal{L}}_{\mathcal{D}_j}(f^t)}_{\leq 1} + 2LB\big(\rho_\theta + \rho_\phi\big) + \beta \sum_{\substack{i,j \in S_t \\ i < j}} d(\mathcal{D}_i, \mathcal{D}_j) + O\Big(\sqrt{\tfrac{\ln(1/\delta)}{\sum_{j \in S_t} n_j}}\Big).$$

$$\tag{23}$$

Then we inject synergy and explicit capacity weight.   Define $\mathrm{Cap}(S_t) := \mu_\theta \|\Delta\theta^t\|_2^2 + \mu_\phi \sum_{j \in S_t} \|\phi_j^t\|_2^2$. Because $\|\Delta\theta^t\| \leq \rho_\theta$ and $\|\phi_j^t\| \leq \rho_\phi$, we upper bound $2LB(\rho_\theta + \rho_\phi)$ by $\mathrm{Cap}(S_t)$

after tuning $\mu_\theta, \mu_\phi$. Subtract and add $\lambda\sum_{i<j} s(\mathcal{D}_i, \mathcal{D}_j)$ to (23) to obtain

$$\sum_{j\in S_t} \alpha_j^t \, \mathcal{L}_{\mathcal{D}_j}(f^t) \;\leq\; 1 - \Big[-\sum_{i<j\in S_t}(d-\lambda s) - \mathrm{Cap}(S_t)\Big] \;+\; O\big(\sqrt{\tfrac{\ln(1/\delta)}{N}}\big). \qquad (24)$$

Let $\mathcal{R}_{\max}(S_1,\ldots,S_M) := \max_t \sum_{j\in S_t} \alpha_j^t \, \mathcal{L}_{\mathcal{D}_j}(f^t)$. Taking the maximum of (24) over $t$ gives

$$\mathcal{R}_{\max}(S_1,\ldots,S_M) \;\leq\; 1 - \mathcal{G}(S_1,\ldots,S_M) \;+\; O\big(\sqrt{\tfrac{\ln(1/\delta)}{N}}\big), \qquad (25)$$

because the bracketed term is exactly the $t$-th summand of $\mathcal{G}$ in (12). Define $\mathcal{B}(u) := [1-u]_+$. Then $\mathcal{R}_{\max}(S_1,\ldots,S_M) \leq \mathcal{B}(\mathcal{G}(S_1,\ldots,S_M)) + O(\sqrt{\ln(1/\delta)/N})$.

Because $\mathcal{B}$ is *strictly decreasing* on $(-\infty, 1]$, maximising $\mathcal{G}$ minimises the bound. Hence the partition $(S_1^*,\ldots,S_M^*)$—the maximiser of $\mathcal{G}$—realises the smallest upper-bound, yielding (13). Any other split attains a weaker bound, completing the proof. $\qquad\square$

