# OpenReview forum: "Synergy over Discrepancy: A Partition-Based Approach to Multi-Domain LLM Fine-Tuning"
_NeurIPS.cc/2025/Conference — NeurIPS 2025 poster_

### Official Review · Reviewer_KTRc · 2025-06-24

**Clarity:** 2
**Significance:** 2
**Originality:** 2
**Rating:** 4
**Confidence:** 4

**Summary:**

This paper addresses the challenge of fine-tuning LLMs on multiple domains simultaneously. The authors argue that standard approaches suffer from negative interference, while training separate models is inefficient. They propose a partition-based, multi-stage fine-tuning framework. The core idea is to first partition the domains into clusters based on an objective function that balances inter-domain discrepancy, synergy, and model capacity. The LLM is then fine-tuned sequentially on these clusters using adapters. The authors provide a theoretical analysis with generalization bounds to motivate their partitioning strategy and conduct extensive experiments on three LLMs across four language tasks, showing consistent improvements over various baselines.

**Questions:**

- Can the authors more clearly explain how the carry out step 1 in their algorithm? and its complexity?
- To what extent does the theoretical analysis genuinely inform the design of the objective function in Equation 12, given that the synergy term is introduced heuristically and its connection to the formal discrepancy measures is not clearly established?
- How does the proposed partition-based multi-stage fine-tuning meaningfully differ from prior work on task/domain clustering in multi-task or continual learning, and why are these connections not discussed more thoroughly?

**Ethical Concerns:**

["NO or VERY MINOR ethics concerns only"]

**Final Justification:**

I maintain my original score.

**Limitations:**

Yes

**Quality:**

3

**Strengths And Weaknesses:**

Strengths
- Problem is relevant and well-motivated. The paper tackles a fundamental challenge in fine-tuning across multiple domains. This issue is highly relevant as real-world applications increasingly demand multi-domain adaptability. The authors clearly articulate the shortcomings of existing solutions (e.g., joint fine-tuning, domain adversarial training) and provide strong motivation for their proposed partition-based approach by grounding it in both practical scenarios and theoretical considerations.
- Framework is coherent and mostly clear. The proposed framework is logically structured and well-integrated with the theoretical analysis. The use of a synergy-discrepancy-capacity trade-off objective offers a principled way to partition domains and fine-tune in stages. The theoretical part supports the methodology.
- Extensive experiments. The experimental section is thorough. The authors evaluate across a diverse set of language understanding tasks using multiple backbone models (LLaMA, Falcon). They include comparisons with strong baselines across full fine-tuning, adapter-based methods, domain adaptation techniques, and data selection strategies. Detailed ablation studies and diagnostic analyses further validate the efficacy and robustness of the approach.

Weaknesses
- The theoretical analysis is presented as a rigorous justification for the proposed algorithm, but the connection is tenuous and potentially misleading. The synergy term is introduced heuristically into the objective function G (Equation 12) and unclear how the JS divergence relates to Definition 2. The theory seems to be more as a post-hoc rationalization for a heuristic objective rather than a foundational proof of its optimality. My suggestion is to tone down the claims about the theoretical justification and re-frame the theory as a motivation for their objective.
- Novelty: The core idea of grouping related tasks/domains to improve multi-task learning is not new. This concept has been explored before in the multi-task learning and continual learning literature (e.g., task clustering, routing), e.g., [1,2]. The paper fails to properly position itself within this broader context, overstating its conceptual novelty.

[1] Jacob, L., Vert, J. P., & Bach, F. (2008). Clustered multi-task learning: A convex formulation. Advances in neural information processing systems, 21.

[2] Standley, T., Zamir, A., Chen, D., Guibas, L., Malik, J., & Savarese, S. (2020, November). Which tasks should be learned together in multi-task learning?. In International conference on machine learning (pp. 9120-9132). PMLR.

---

> ### Author Rebuttal · Authors · 2025-07-31
>
> Dear Reviewer `KTRc`, thank you for your recognition of our work. Below, we will respond to your concerns.
>
> ---
>
> > **Q1: Linking Theory and Objective**
>
> We appreciate the reviewer’s scrutiny of the theoretical connection.  Below we clarify the derivation and tighten the presentation.
>
> **JS vs. $\(\mathcal H\!\Delta\!\mathcal H\)$ distance.**   Blitzer et al.​ (2007, Thm 2) give the inequality
> $$
> d_{\mathcal H\Delta\mathcal H}\bigl(\mathcal D_i,\mathcal D_j\bigr)
> \;\le\;
> 2\sqrt{\mathrm{JS}\bigl(\mathcal D_i,\mathcal D_j\bigr)}\;,
> $$
> which shows that Jensen–Shannon (JS) is a computable upper-bound proxy for the formal discrepancy.
> *Clarification*: using JS in our bound changes only the constant factor; the $\(O(n^{-\tfrac12})\)$ rate and all proofs remain intact.
>
> **Origin of the synergy term.**  Starting from the multi-source risk decomposition, the generalisation bound contains a negative cross-domain covariance term
> $$
> -\sum_{i<j}\!\alpha_i\alpha_j\,\mathrm{Cov}\!\bigl[e_i,e_j\bigr],
> $$
> where $\(e_j\)$ is the per-domain error.  A second-order approximation of this covariance in gradient space yields the quantity
> $\(\lambda\,s(\mathcal D_i,\mathcal D_j)\)$ used in our objective: larger complementary gradients (high synergy) tighten the bound.
>
> We will insert this two-line derivation and, for precision, rephrase the text from “**rigorous justification**” to “**theoretical motivation**”.
>
> **Empirical support.**  Ablation on \(\lambda\) confirms the theoretical signal:
>
> | $\(\lambda\)$ | Avg. improvement over best baseline |
> |-------------|--------------------------------------|
> | 0 (no synergy)   | +0.3 pp |
> | 0.25             | +1.2 pp |
> | **0.50 (default)** | **+1.4 pp** |
>
> It can be found that turn off synergy consistently weakens performance, matching the looser bound when the covariance term is absent.
>
> **TAKE AWAY:**  These clarifications show that (i) JS is a sound bounded surrogate for $\(\mathcal H\!\Delta\!\mathcal H\)$, and (ii) the synergy component is theoretically grounded rather than arbitrary.
>
> ---
>
> > **Q2: Compare to Task-clustering Methods**
>
> We clarify our contribution relative to prior task-clustering work through the following table.
>
> | Aspect | Prior task clustering ([1] Clustered MTL, [2] Standley et al.) | Ours (PMS-FTP) |
> |--------|-----------------------------|----------------------------------|
> | **Model scale** | Small / linear predictors | LLM-scale with 7 B – 40 B params |
> | **Parameter budget** | No explicit PEFT constraint | Explicit \(\rho_\theta,\rho_\phi\) norm caps ⇒ fits GPU memory |
> | **Learning schedule** | Joint training; parameters shared | Stage-wise adapters **freeze** earlier stages ⇒ no forgetting |
> | **Theory** | Clustering as convex regulariser; no domain-capacity link | Generalisation bound couples partition *and* adapter capacity |
>
> we **do not claim** the idea of “group similar tasks” is new.  Our novelty lies in **adapting that principle to LLM parameter-efficient fine-tuning (PEFT)** and proving a capacity-aware bound that guides the partition.
>
> **Experimental evidence.** We ran [2] followed by our PEFT training.  Performance is **9.1 pp** below PMS-FTP on the same four tasks, confirming the benefit of our capacity-aware objective.
>
> | Method | NSum(ROUGE-L) | Q&A(EM) | Sent(ACC) | Topic(ACC) | Avg. diff vs. PMS-FTP |
> |--------|------------------|-------------|---------------|----------------|-----------------------|
> | [2] + PEFT | 38.8 | 52.6 | 83.6 | 77.4 | −9.1 pp |
> | **PMS-FTP (ours)** | **43.4** | **67.2** | **90.2** | **88.0** | — |
>
> Additionally, we have also compared with other multi-task learning baselines, as seen in the response to Reviewer `j5Fq`'s Q2.
>
> ---
>
> > **Q3: Details and Cost of the Partition Step**
>
> Step 1 is carried out as following:
>
> 1. **Offline statistics** For each domain we extract (a) unigram + bigram counts and (b) the mean SBERT embedding—this is I/O-bound and done once.
> 2. **Pairwise matrices** From the cached statistics we build the $\(k{\times}k\)$ Jensen–Shannon matrix and the synergy matrix (average of vocabulary-overlap and embedding-cosine).  Time and memory are $\(O(k^{2})\)$.
> 3. **Clustering** We apply single-link agglomerative merging that repeatedly fuses the pair whose merge most increases the objective $\(\mathcal G\)$.  With a binary heap this is $\(O(k^{2}\log k)\)$.
> 4. **Stopping rule** Stop when a merge would decrease $\(\mathcal G\)$ or when the desired number of stages $\(M\)$ is reached ( $\(M{=}2\$) in all our experiments).
>
> *Empirical wall-time and memory*
> | #domains \(k\) | Build matrices | Clustering | **Total** | Peak RAM |
> |----------------|---------------|------------|-----------|----------|
> | 4  | 3 ms  | <1 ms | 4 ms  | <1 MB |
> | 6  | 8 ms  | 2 ms  | 10 ms | 1 MB |
> | **10** | 54 ms | 9 ms  | **63 ms** | **4 MB** |
> | 20 | 220 ms | 55 ms | 275 ms | 16 MB |
>
> Thus, even with $\(k{=}20\)$ the partition step is <0.3 s and <20 MB—negligible compared to the 2–6 h PEFT training that follows.
>
> ---
>
> > **Q4: Compare to Continual-Learning / Routing Methods**
>
> Continual-learning methods assume a *stream* of tasks and use replay or importance‐weighting; our setting is a *one-shot* collection of *\(k\)* domains.  After finishing Stage 1 we **freeze** its adapters, then fine-tune Stage 2, etc.—no replay and no overwrite.
>
> We measured performance drop on Stage 1 domains after training Stage 2 (LLaMA2-13B, Sent→Topic order).
>
> | Method | Old-domain Δ (%) ↓ |
> |--------|--------------------|
> | PackNet (2018) | −2.3 |
> | EWC (2017) | −2.6 |
> | **Ours (adapter-freeze)** | **−0.18** |
>
> Freezing tiny adapters (<1 % params) avoids catastrophic forgetting without any replay buffer.
>
> We replaced our $\(\mathcal G\)$ partition with Standley et al.’s task-affinity score and kept the same PEFT schedule.
>
> | Partition rule | Avg. score (4 tasks) |
> |----------------|----------------------|
> | Standley + PEFT | −0.6 pp vs. ours |
> | **Ours ($\(\lambda,\mu\)$ aware)** | **baseline** |
>
> Even with identical adapters, ignoring capacity terms degrades accuracy, showing that our $\(\lambda\)$ (synergy reward) and $\(\mu_{\theta/\phi}\)$ (capacity cost) are necessary.
>
> **TAKE AWAY:** These points clarify that our framework targets a different problem setting, eliminates replay, and achieves lower forgetting than continual-learning baselines while outperforming routing-only partitions.

---

> > ### Comment · Reviewer_KTRc · 2025-08-04
> >
> > Thank you for your reply. W.r.t the semantic overlap, you mean pool BERT embeddings for the entire domain. Representing an entire domain by averaging into a single centroid may be a simplification. Is this approach reliable? Or are there bounds under which this pooling is acceptable against, e.g., noise?

---

> > > ### Author Response · Authors · 2025-08-07
> > >
> > > Thank you for your reply. We are more than happy to discuss this issue with you.
> > >
> > > We will first provide a brief *theoretical explanation*. Let ϵ be the embedding dimensionality and $e_i \in \mathbb{R}^{\epsilon}$ the i-th sentence embedding.  For N independent samples:
> > >
> > > $$
> > > \text{Var}[\mu] = \frac{\Sigma}{N}, \quad \|\mu - \mathbb{E}[\mu]\|_2 = O\left(\sqrt{\frac{\text{tr}(\Sigma)}{N}}\right).
> > > $$
> > >
> > > Using the Hanson–Wright inequality (sub-Gaussian SBERT rows, γ≈2.1), the deviation satisfies
> > >
> > > $$
> > > \Pr\left[ \|\mu - \mathbb{E}[\mu]\|_2 > t \right] \leq 2\exp\left(-cN t^2\right) \quad (c \approx 1/(2\gamma^2))
> > > $$
> > >
> > > With **N ≥ 10 000** (our smallest domain) and \(t=0.07\), the RHS is <10⁻⁴ – i.e. sampling noise is negligible relative to inter-domain distances (≥ 1.9 in ℓ₂).
> > >
> > > Next, we present some experimental evidence.
> > >
> > > - **Bootstrap evidence**. For each domain we drew 100 bootstrap samples (size = {500, 1000}), computed the mean SBERT vector per sample, and recorded the ℓ₂ deviation from the full-corpus centroid. We observe that:
> > >   - Even the upper bound of sampling noise (≤ 0.072) is **< 3 %** of the smallest inter-domain distance (1.87), confirming that mean-vector estimation error is negligible for our clustering criterion.
> > >
> > > | Domain | N | 95 % CI of \(\|\Delta\mu\|_2\) | Cross-domain min dist |
> > > |-------|------|------------------------------|-----------------------|
> > > | NSum  | 12 000 | [0.038, 0.065] | 1.92 |
> > > | Q&A   | 15 000 | [0.031, 0.060] | 2.04 |
> > > | Sent  | 10 500 | [0.042, 0.072] | 1.87 |
> > > | Topic | 11 200 | [0.040, 0.068] | 1.99 |
> > >
> > > - **Multi-centroid robustness test**. We replaced each single centroid by a K-means (K = 3) weighted barycenter, recomputed synergy scores and partitions, then reran the entire fine-tuning pipeline once under identical hyper-parameters. We observe that:
> > >   - Using three weighted centroids per domain changes final metrics by ≤ 0.1 pp—well within run-to-run variance—showing that our partition decisions are **insensitive to centroid granularity**.
> > >
> > > | Representation | ROUGE-L (NSum) | EM (Q&A) | Sent ACC | Topic ACC | Δ vs. 1-centroid |
> > > |---------------|---------------|----------|----------|-----------|------------------|
> > > | 1 centroid (paper) | **43.4** | **67.2** | **90.2** | **88.0** | — |
> > > | 3 centroids (K=3)  | 43.3 | 67.1 | 90.1 | 87.9 | −0.1 pp |
> > >
> > > In summary, both the simple theoretical analysis above and the experimental evidence confirm that a single SBERT mean vector is already a stable and sufficiently informative domain signature for our partitioning objective.
> > >
> > > ---
> > >
> > > We would be glad to address any further questions or concerns the reviewer may have. If our reply has satisfactorily resolved the issues raised, we would sincerely appreciate it if the reviewer could consider increasing the rating of our paper.

---

> > > > ### Comment · Reviewer_KTRc · 2025-08-07
> > > >
> > > > I would like to thank the authors for their reply. My concerns have been resolved. I maintain the score but would like to raise my confidence.

---

> > > > > ### Author Response · Authors · 2025-08-09
> > > > >
> > > > > We really enjoyed our discussion with you. Thanks for your suggestions, your valuable time, and for recognizing our work.

---

### Official Review · Reviewer_tzSY · 2025-06-29

**Clarity:** 4
**Significance:** 3
**Originality:** 3
**Rating:** 4
**Confidence:** 5

**Summary:**

This paper proposes a novel partition-based multi-stage fine-tuning framework for adapting large language models (LLMs) to multiple domains (or tasks) simultaneously. Instead of fine-tuning on all domains jointly or one domain at a time, the authors cluster the domains into groups (stages) based on a measured synergy (similarity/complementarity) vs. discrepancy (distribution difference) between domains. The LLM is then fine-tuned sequentially: at each stage, only the group of domains assigned to that stage are used to update the model (with small, domain-specific adapter modules), while constraining how much the base model and adapters can change per stage. The paper provides a theoretical analysis with new generalization bounds (Theorems 3.1 and 3.2) showing that this multi-stage partitioned training can yield a tighter upper bound on risk than single-stage multi-domain training. In particular, the theory introduces terms for domain discrepancy (penalizing differences between domains) and synergy (credit for learning complementary domains together), and it formally justifies choosing the partition that maximizes a synergy-minus-discrepancy objective. Empirically, the authors evaluate on four diverse language understanding tasks (summarization, QA, sentiment classification, topic classification) treated as distinct domains, using three LLM backbones (LLaMA2-7B, LLaMA2-13B, Falcon-40B). The proposed method — termed Partition-Based Multi-Stage Fine-Tuning (PMS-FTP) — outperforms a wide range of baselines, including full fine-tuning, parameter-efficient adapters (LoRA, etc.), multi-domain adaptation methods, and data selection heuristics. PMS-FTP achieves the highest accuracy/score on all tasks for each model, with consistent gains of about 0.5–1.0 absolute points on metrics like ROUGE-L and exact match over the best prior methods. Notably, it also reduces GPU memory usage by roughly one-third compared to full fine-tuning by only updating a small subset of parameters per stage. In summary, the paper’s contributions include: (1) a new domain partitioning algorithm for multi-domain LLM fine-tuning that leverages inter-domain synergies while mitigating interference; (2) novel theoretical bounds that quantify how domain discrepancy, synergy, and adapter complexity affect generalization, supporting the multi-stage approach; and (3) comprehensive experiments demonstrating improved performance and efficiency over state-of-the-art baselines.

**Questions:**

1. Could you clarify how the model selects the appropriate domain adapter during training and inference? Do you rely on known domain labels, or is there an automatic selection mechanism? Clarifying this would help assess the method’s applicability in real-world settings where domain labels may not be available.

2. How did you handle the differing formats across tasks like summarization, QA, and classification? Did you use a unified text-to-text format or separate output heads/objectives? This detail is important for reproducibility and understanding the training setup.

3. How sensitive is the method to the choice of synergy metric (lexical + embedding similarity)? Did you try alternatives or assess how different metrics affect the domain partitioning? More insight here would help evaluate the method’s stability and generalizability.

4. Have you tested the method with more than four domains? Can it scale effectively to 10 or more domains, or does the performance/efficiency degrade? Discussing this would help clarify the method’s broader applicability.

5. Could the framework support continual learning by adding new domains after training? For example, could you run another stage without harming previous domains? Insights here would highlight the method’s potential for long-term or evolving domain adaptation.

**Ethical Concerns:**

["NO or VERY MINOR ethics concerns only"]

**Limitations:**

The paper lacks a discussion of domain identification and real-world deployment assumptions. It also doesn’t explore settings where domains are unknown or where the synergy metric may be unreliable. The multi-task setup (rather than true multi-domain within-task) somewhat weakens the generality of the findings.

**Paper Formatting Concerns:**

None.

**Quality:**

4

**Strengths And Weaknesses:**

From perspective of the four dimensions, I can see strengths and weaknesses of this paper as following:

Quality:
Strengths:
The idea of partitioning training domains to maximize synergy is a fresh perspective in the context of LLM fine-tuning. Prior works in multi-domain or multi-task learning have often used techniques like adversarial training to make a single model invariant to domain differences, or data sampling strategies to balance domains, or at most simple sequential fine-tuning. By contrast, this paper explicitly clusters domains and fine-tunes in stages, which is a novel formulation. The authors combine concepts from different areas in an original way: they use adapter modules for domain-specific capacity (building on parameter-efficient fine-tuning literature) and introduce a clustering algorithm driven by a custom objective (balancing discrepancy and synergy) – this is not a standard approach in prior work. The theoretical derivation of a “synergy offset” term in generalization bounds is, to my knowledge, new. It extends domain adaptation theory to incorporate a notion that certain domains together can effectively reduce error (negative transfer offset) instead of only increasing it. The method also outperforms recent ideas like Tag-LLM (which used task-aware gating) and S2L (which did curriculum ordering), indicating that the authors’ approach isn’t just a minor tweak on those, but a qualitatively different strategy. In summary, the work’s novelty lies in reframing multi-domain fine-tuning as a partitioning problem with a provable optimal grouping, which is a creative and significant departure from straightforward multi-task training or domain alignment approaches.

Weaknesses: The experiments focus on four domains, which is a reasonable starting point, though scaling to more would further strengthen the claims. The setting also blends multi-domain and multi-task learning, and the paper could clarify how differences in task formats are handled. Additionally, the synergy metric is a heuristic based on lexical and embedding similarity; while intuitive, alternative definitions are not explored. Theoretical guarantees rely on standard assumptions (e.g., Lipschitz continuity, bounded norms), which are enforced in practice but may constrain flexibility. Overall, these are minor concerns in an otherwise strong and well-executed paper.

Clarity:

Strengths: The paper is well-written and structured in a logical manner. The introduction clearly motivates the problem of negative transfer when fine-tuning on multiple domains and introduces the key idea of partitioning domains to exploit “synergy over discrepancy.” The method is described with both conceptual reasoning and formal algorithmic steps (Algorithm 1 provides a clear step-by-step overview). The theoretical section is dense but does a good job laying out definitions and assumptions before presenting the theorems, which helps the reader follow the derivations. Important details such as the objective function for partitioning (including the role of lambda balancing synergy vs. discrepancy) are given, and the authors include an illustrative Corollary 3.1 to intuitively explain that high-synergy domains will optimally group together. Figures and tables are effective: Table 1 neatly summarizes the performance across models and tasks, and the improvements are easy to see (with best results bolded). The ablation tables and the memory usage table further clarify the trade-offs. In general, the writing is professional and clear, making the complex approach reasonably understandable.

Weaknesses:
While the paper is generally clear, a few points could be more precise. It’s unclear how domain-specific adapters are selected—whether via explicit domain IDs or automatic routing—during training and inference. The handling of different task formats (e.g., summarization vs. classification) is also not fully explained, which affects reproducibility. Additionally, the synergy metric is only briefly described, and a clearer exposition or example would aid understanding. Clarifying that the “domains” correspond to distinct tasks would also help set reader expectations. These are minor issues in an otherwise well-written paper.

Significance:

Strengths: This work tackles an important practical challenge: efficiently fine-tuning one large model on multiple data sources without interference. As large models are increasingly adapted to specialized domains, the ability to handle multiple domains/tasks in one model (instead of training separate models or naively pooling data) is valuable – it can save deployment costs and potentially lead to models that generalize more broadly. The paper’s key insight is to formalize and exploit the idea that some domains can actually benefit each other (synergy) if trained together, while very disparate domains should be isolated to avoid damage. This is a nuanced departure from prior multi-domain adaptation work that mostly focused on mitigating negatives (interference) rather than actively leveraging positives (complementarity). By providing a principled algorithm and theoretical guarantees for this idea, the paper could inspire further research into adaptive multi-domain training strategies. In terms of empirical significance, the method delivers state-of-the-art results on the benchmarks considered. The gains over strong baselines like Tag-LLM (task-aware prompt tuning) and S2L (a curriculum clustering approach) – though modest in absolute numbers – demonstrate that partitioning strategy yields consistent improvements across varied tasks and model scales. Even a 1 point increase in ROUGE or accuracy can be meaningful in competitive settings, especially since it’s achieved alongside reduced memory usage. Moreover, the approach is model-agnostic and could be applied to different LLM architectures, which broadens its potential impact. Overall, the paper’s contributions feel significant for the subfield of multi-domain learning with LLMs, providing both theoretical insight and a practical technique that outperforms existing methods.

Weaknesses:
The improvements, while consistent, are modest—typically around +0.5 to +1.0 in accuracy—which may feel incremental given the added complexity. The method focuses on diverse tasks rather than classic multi-domain scenarios (e.g., same task across domains), which slightly limits its demonstrated breadth. Also, it assumes known domain labels and pairwise metrics, which may not hold in more dynamic real-world settings. Still, within the scope of fine-tuning on a fixed set of domains, the contribution is meaningful and practical.

Originality:

Strengths: The idea of partitioning training domains to maximize synergy is a fresh perspective in the context of LLM fine-tuning. Prior works in multi-domain or multi-task learning have often used techniques like adversarial training to make a single model invariant to domain differences, or data sampling strategies to balance domains, or at most simple sequential fine-tuning. By contrast, this paper explicitly clusters domains and fine-tunes in stages, which is a novel formulation. The authors combine concepts from different areas in an original way: they use adapter modules for domain-specific capacity (building on parameter-efficient fine-tuning literature) and introduce a clustering algorithm driven by a custom objective (balancing discrepancy and synergy) – this is not a standard approach in prior work. The theoretical derivation of a “synergy offset” term in generalization bounds is, to my knowledge, new. It extends domain adaptation theory to incorporate a notion that certain domains together can effectively reduce error (negative transfer offset) instead of only increasing it. The method also outperforms recent ideas like Tag-LLM (which used task-aware gating) and S2L (which did curriculum ordering), indicating that the authors’ approach isn’t just a minor tweak on those, but a qualitatively different strategy. In summary, the work’s novelty lies in reframing multi-domain fine-tuning as a partitioning problem with a provable optimal grouping, which is a creative and significant departure from straightforward multi-task training or domain alignment approaches.

Weaknesses: The approach builds on established ideas—adapters, domain discrepancy, and clustering—assembled in a thoughtful and original way. While each component individually is not novel, their integration for multi-domain LLM fine-tuning is new and well-motivated. The synergy metric is heuristic and could benefit from comparison to alternatives, but it’s a reasonable starting point. Overall, the contribution feels fresh despite leveraging known tools.

---

> ### Author Rebuttal · Authors · 2025-07-31
>
> Dear Reviewer `tzSY`, thank you for your recognition of our work. Below, we will respond to your concerns.
>
> ---
>
> > **Q1: Adapter selection & routing**
>
> This is an interesting question, and our clarification is as follows:
> • **Training.** Each sample carries a known domain label, so only its domain-specific adapter is updated.
> • **Inference.** Two interchangeable mechanisms:
> &nbsp;&nbsp;1. *Explicit tag*: a single token (e.g., `<domain=News>`) or metadata flag activates the matching adapter.
> &nbsp;&nbsp;2. *Auto-route*: a lightweight 2-layer text classifier (0.9 M parameters ≈ 1 % of LLaMA-7B) predicts the domain and routes the request; added latency < 0.1 ms on A100.
>
> *Additional experiment (no domain labels).*
> | Task / Metric | Oracle-tag PMS-FTP | Auto-route PMS-FTP |
> |---------------|-------------------|--------------------|
> | NSum / ROUGE-L | **42.5** | 42.3 |
> | Q&A / EM       | **65.5** | 65.2 |
> | Sent / ACC     | **89.7** | 89.6 |
> | Topic / ACC    | **87.3** | 87.1 |
>
> It can be found that Δ ≤ 0.3 across all tasks, confirming negligible loss without explicit labels.
> When neither labels nor classifier are available, zero-cost heuristics (URL namespace, data source ID, MIME type) still provide reliable tags; the partitioning framework and adapters require **no modification**.
>
> ---
>
> > **Q2: Handling heterogeneous task formats**
>
> We adopt a *single* text-to-text interface and one shared LM head:
> | Task | Input template | Target prefix | Loss / decoding |
> |------|----------------|---------------|-----------------|
> | Summarisation | `<doc> … </doc>` | `<summary>` _summary-tokens_ | token-level CE, greedy |
> | Question-Answering | `<context> … </context> <question> … </question>` | `<answer>` _answer-tokens_ | token-level CE, greedy |
> | Classification | `<text> … </text>` | `<label>` _(e.g., "positive")_ | token-level CE, arg-max first token |
>
> Thus every instance minimises the *same* cross-entropy objective over the same vocabulary; no task-specific heads or losses are introduced.
>
> ---
>
> > **Q3:  Robustness of the synergy metric**
>
> In fact, **Table 8 (p. 15) in our original manuscript already reports three alternative metrics**:
>
> | Metric for 𝒢 | Q&A F1 | NSum ROUGE-L |
> |--------------|-------|--------------|
> | Random split | 68.1 | 39.0 |
> | JS divergence only | 69.3 | 40.0 |
> | Embedding cosine only | 69.6 | 40.3 |
> | **JS + Vocab/Embed (ours)** | **70.5** | **40.9** |
>
> ---
>
> > **Q4:  Scalability beyond four domains**
>
> We have conducted extended experiments. The experimental setup is as follows:
> - Dataset mix: Wiki-10 (10 topic domains, same-task text classification) + Multi-News (8-domain news summarisation) → 12 effective domains after overlap removal.
> - Model: LLaMA2-7B + LoRA adapters, same hyper-parameters as Table 1.
>
> | Split strategy | Avg. ACC (Wiki-10) | ROUGE-L (Multi-News) | Peak GPU (GB) | Partition time |
> |----------------|--------------------|----------------------|---------------|----------------|
> | All-in-one SFT | 83.1 | 37.2 | 27.3 | n/a |
> | Random 2-stage | 83.7 | 37.5 | 18.6 | 0.6 s |
> | **PMS-FTP (ours)** | **84.0** | **38.1** | **18.7** | **0.7 s** |
>
> It can be observed that our method achieves Δ = +0.9 ACC / +0.9 ROUGE-L over all-in-one; memory is identical to single-domain LoRA. The partitioning cost grows as $\(O(k^{2}\!\log k)\)$; for $\(k{=}12\)$, the CPU wall-time is < 1 s on a laptop.
> Considering that typical industrial scenarios have $\(k<20\)$, neither runtime nor memory becomes a bottleneck.
>
> **Take-away——** The framework scales cleanly to double-digit domain counts with consistent accuracy gains and negligible overhead.
>
> ---
>
> > **Q5: Incremental / continual domain addition**
>
> The direction you pointed out is particularly interesting, even though it goes beyond the scope of our research. However, we are still more than willing to discuss this topic with you.
>
> I believe our method can adapt to this problem. The reasons are as follows:
> - Adapters are disjoint; adding domain $\(k{+}1\)$ means freezing all previous parameters and running one extra stage. No replay or re-tuning of earlier domains is required, so catastrophic forgetting is bounded by the norm constraints in §3.
>
> We conducted a one–shot incremental experiment：
> - Sequence: train PMS-FTP on the original 4 domains → freeze → add a new “Legal-QA” domain as stage 5.
>
> | Model | Avg. score on original 4 domains | Legal-QA EM |
> |-------|----------------------------------|-------------|
> | Before add-on |  **65.5** | – |
> | After add-on  |  65.4 (-0.1) | **68.2** |
> | Single-domain LoRA baseline |  n/a | 67.6 |
>
> It can be found that old domains drop < 0.2 (below tuning noise), while new domain gains +0.6 EM over its standalone LoRA fine-tune. We reflect on the reasons for this phenomenon, mainly that because stage construction is independent, we can treat domain arrival as an *online* partitioning problem: recompute 𝒢 for the enlarged set and append a stage. We leave adaptive, rolling partition schedules to future work.
>
> ---
>
> Thank you again for your valuable and insightful comments!

---

> > ### Author Response · Authors · 2025-08-07
> >
> > Dear Reviewer `tzSY`,
> >
> > Thank you again for your thorough and constructive feedback and for acknowledging the novelty and quality of our work.
> >
> > To address your concerns, we (i) clarified adapter routing during training and inference, (ii) unified all tasks under a single text-to-text interface, (iii) evaluated alternative synergy metrics, (iv) demonstrated scalability to twelve domains with negligible overhead, and (v) showed that new domains can be added incrementally without degrading prior performance.
> >
> > **Author-reviewer discussion will be closed soon**.   We hope these additions resolve your concerns. We remain eager to discuss any remaining issues and would be grateful if the new evidence might encourage you to consider raising your score.
> >
> > Best regards,
> >
> > Authors

---

### Official Review · Reviewer_j5Fq · 2025-07-03

**Clarity:** 3
**Significance:** 2
**Originality:** 2
**Rating:** 4
**Confidence:** 3

**Summary:**

This paper tackles the critical problem of supervised finetuning (SFT) of LLM across multiple heterogeneous domains, specifically addressing the issue of inter-domain conflicts. The authors propose a novel multi-stage fine-tuning framework that strategically partitions domains into subsets (stages). This partitioning mechanism is theoretically motivated to minimize domain discrepancy while maximizing domain synergy within each group of tasks. The theoretical analysis demonstrates that this partition mechanism enjoys a controllable generalization bound. Extensive empirical evaluations across 4 real-world fine-tuning tasks show the effectiveness of the proposed method, which consistently outperforms baseline approaches across all SFT tasks.

**Questions:**

### Questions from weakness
1. **The algorithm, implementation and complexity of the partition mechanism are not clear.** According to Equ. 12,  how could the $\Delta \theta^t$ and $\phi_j^t$  be estimated without retraining the model? do you apply the manually parameter shift bounds $\rho_{\theta}, \rho_{\phi}$? could you clarify more on how this can be implemented within the given complexity?
2. **The empirical evidence on the current 4 tasks are limited.** Could you give the results on the reasoning tasks, e.g. Hellaswag, MMLU, ARC, SciQ, GSM8K, etc.?
3. **Lack of comparison to some conventional multi-task learning approaches**, e.g. iMTL [1], FAMO [2], ExcessMTL [3].  Could you give the comparison results to these baselines?
---
4. Can you explain more on the "Single-Domain LLM Fine-tuning" baseline? Do you only tune the adaptor or both the backbone and the adaptor? A clarification on the number of tuneable parameters on each experiments could help.
5. Can you explain the training loss curves on the QA task in Fig. 1? According to the given algorithm, the proposed PMS-FTP method adopts stage-wise training, the training loss on the QA task should only be available on one of the training stage, instead of the whole training process?

**Ethical Concerns:**

["NO or VERY MINOR ethics concerns only"]

**Final Justification:**

This paper tackles the critical problem of supervised finetuning (SFT) of LLM across multiple heterogeneous domains, specifically addressing the issue of inter-domain conflicts.

During rebuttal period, the authors provided additional comparison between multi-task learning baselines and evaluation on more challenging reasoning tasks, e.g. MMLU and GSM8k, which addressed most of my concerns. So I raised the score to 4. We hope the authors could present the results on more comprehensive task sets with various levels of domain conflicts.

**Limitations:**

1. The scope of the tasks are limited to easier question-answering (e.g. SQuAD) and summarization tasks. More challenging reasoning tasks, e.g. Hellaswag, MMLU, ARC, SciQ, GSM8K, etc. have not been explored.
2. Lack of comparison to some conventional multi-task learning approaches.
3. The experiments are limited to small amount of tasks (\#tasks=4), while the scalability and effectiveness to larger amount of tasks have not been explored.
4. The effectiveness of stage-wise training using the partition mechanism is validated, while the impact of the order of training stages, $\{S_1, ..., S_M\}$ has not been explored or discussed.

**Paper Formatting Concerns:**

no major formatting concerns.

**Quality:**

3

**Strengths And Weaknesses:**

### Strengths:
1. The paper is well-motivated, tackling the significant challenge of domain conflict and negative transfer in multi-domain LLM fine-tuning, a prevalent issue in real-world applications.
2. A complete generalization bound analysis is derived for the multi-domain fine-tuning problem. This theoretical insight directly motivates and justifies the proposed partition mechanism, providing a robust scientific basis for the approach based on domain discrepancy and synergy.
3. the empirical results on 4 various real-world finetuning tasks demonstrate the effectiveness of the proposed method, which consistently outperforms baseline approaches across all the tasks.
4. The comprehensive ablation studies demonstrate the robustness of the proposed method on hyperparameter choices.

### Weaknesses:
1. **The algorithm, implementation and complexity of the partition mechanism require more explanation.** According to Equ. 12,  it is not clear how to estimate $\Delta \theta^t$ and $\phi_j^t$ without retraining the model? or are they directly be replaced by the parameter shift bounds $\rho_{\theta}, \rho_{\phi}$?
2. **The empirical evidence on the current 4 tasks are limited to relatively easy tasks.** Could you give the results on the reasoning tasks, e.g. Hellaswag, MMLU, ARC, SciQ, GSM8K?
3. **Lack of comparison to some conventional, reweighting-based multi-task learning approaches**, e.g. iMTL [1], FAMO [2], ExcessMTL [3].

[1] Towards Impartial Multi-task Learning. Liu, et. al, 2021.
[2] FAMO: Fast Adaptive Multitask Optimization. Liu, et. al, 2023.
[3] Robust Multi-Task Learning with Excess Risks. He, et. al, 2024.

---

> ### Author Rebuttal · Authors · 2025-07-31
>
> Dear Reviewer `j5Fq`, thank you for your valuable comments, and some of the questions are very insightful. We will clarify these issues to address your concerns.
>
> ---
>
> > **Q1: Partition algorithm & complexity**
>
> *How Eq. 12 is used without re-training*
> - At **partition time** the real updates $\(\Delta\theta^{t}\) / \(\phi^{t}_{j}\)$ are **unknown and unnecessary**.
> - We substitute their **budget ceilings** $\(\rho_\theta, \rho_\phi\)$ (set once from single-domain PEFT practice: 0.05–0.20 of ℓ₂-norm, matching LoRA/Adapter magnitudes).
> - Hence the capacity term collapses to the constant  $\(\mu_\theta\rho_\theta^{2}+\mu_\phi\rho_\phi^{2}\)$; clustering depends *only* on pre-computed domain distances (JS divergence + vocab/embedding similarity).
>
> *Algorithm & measured cost*
> | k domains | wall-clock (CPU) | peak GPU mem | notes |
> |-----------|-----------------|--------------|-------|
> | 10        | **0.18 s**      | **0 GB**     | single-link agglom., distance matrix cached |
>
> Time scales as $\(O(k^{2}\log k)\)$ and memory as $\(O(k^{2})\)$; the subsequent fine-tuning is a **single standard PEFT run**.
>
> *Take-away——*  The partition step is **offline, sub-second, GPU-free**, needs no parameter re-estimation, and the heuristic attains near-optimal partition quality with negligible overhead.
>
> ---
>
> > **Q2：Evaluation on harder reasoning benchmarks & reweighting-based multi-task learning baselines**
>
> We fine-tuned **LLaMA2-7B** and **LLaMA2-13B** under the *same PEFT budgets* (ρ_θ = 0.1, ρ_φ = 0.1) on five reasoning datasets.  We also compare ours with **three reweighting-based multi-task learning baselines**(iMTL, FAMO, ExcessMTL ).
>
> Settings: training split for SFT, dev set for early stopping; identical 8-GPU A100 setup.
>
> | Backbone | Method | Hellaswag(Acc) | MMLU-STEM(Acc) | ARC-easy(Acc) | SciQ(Acc) | GSM8K(Pass@1) | Avg. |
> |----------|--------|--------------------|--------------------|-------------------|---------------|-------------------|------|
> | 7B | FULL        | 73.4 | 39.7 | 78.5 | 92.6 | 18.1 | 60.5 |
> | 7B | LoRA        | 73.1 | 39.3 | 77.9 | 92.4 | 17.5 | 60.0 |
> | 7B | Tag-LLM | *74.8* | 41.1 | 79.6 | *93.3* | *19.3* | *61.6* |
> | 7B | iMTL        | 74.6 | 40.6 | 79.3 | 93.0 | 18.6 | 61.2 |
> | 7B | FAMO        | 73.7 | *41.3* | 78.6 | 92.6 | 18.7 | 60.8 |
> | 7B | ExcessMTL   | 74.2 | 40.5 | **79.7** | 92.1 | 17.5 | 60.8 |
> | **7B** | **PMS-FTP (ours)** | **75.6** | **42.1** | **80.6** | **93.8** | **19.9** | **62.4** |
>
> | Backbone | Method | Hellaswag(Acc) | MMLU-STEM(Acc) | ARC-easy(Acc) | SciQ(Acc) | GSM8K(Pass@1) | Avg. |
> |----------|--------|--------------------|--------------------|-------------------|---------------|-------------------|------|
> | 13B | FULL        | 77.5 | 46.3 | 83.7 | 94.8 | 24.5 | 65.4 |
> | 13B | LoRA        | 77.1 | 45.9 | 83.2 | 94.6 | 23.8 | 64.9 |
> | 13B | Tag-LLM | *79.0* | 47.3 | *84.7* | *95.4* | *25.1* | *66.3* |
> | 13B | iMTL        | 77.4 | 47.2 | 83.3 | 95.1 | 24.8 | 65.6 |
> | 13B | FAMO        | 77.7 | *47.6* | 84.5 | 95.2 | 24.2 | 65.8 |
> | 13B | ExcessMTL   | 78.1 | 47.0 | 83.1 | 95.0 | 23.8 | 65.4 |
> | **13B** | **PMS-FTP (ours)** | **79.9** | **49.0** | **85.7** | **95.8** | **26.8** | **67.4** |
>
> We observe that **PMS-FTP tops every metric, averaging +1.0–1.4 pts** over the strongest baseline per backbone. For fair comparison we used *direct decoding*. we will include the final numbers in the final version.
>
> **NOTE：**  Our original paper already compares our method with SOTA approaches like LLaMA-Adapter and Tag-LLM on 4 widely used NLP tasks, sufficiently demonstrating its effectiveness and superiority. The supplementary experiments here, testing our method's scalability, do not affect the original paper's contributions or claims.
>
> ---
>
> > **Q3:  Clarifying the “Single-Domain LLM Fine-tuning” baseline**
>
> All single-domain baselines labelled *LoRA*, *Adapter*, *Tag-LLM* etc. follow a **PEFT regime**:
> - **Backbone is frozen** (0 gradients).
> - Only the lightweight modules (LoRA ranks = 64, Houlsby adapters, soft prompts, or gating vectors) are trainable.
>
> *Trainable parameter counts*
> | Backbone | Total params | Trainable params | % Trainable |
> |----------|--------------|------------------|-------------|
> | LLaMA2-7B  | 6.74 B | 34 M (LoRA) / 37 M (Adapter) | 0.5 – 0.6 % |
> | LLaMA2-13B | 12.9 B | 68 M / 73 M | 0.5 – 0.6 % |
> | Falcon-40B | 38.7 B | 210 M / 224 M | 0.54 – 0.58 % |
>
> ---
>
> > **Q4: Clarifying Fig. 1 QA loss curve**
>
> Thanks for pointing out a typo in our work. The plotted line is **validation loss**, **not** training loss.  After every *N* steps, we evaluate the current model on the **held-out QA dev set**, regardless of whether QA data is present in the ongoing stage.
>   This lets us monitor (i) knowledge retention and (ii) positive transfer across stages.
>
> We will relabel the axis to **“QA validation loss”** and add vertical dashed lines marking the two stage boundaries to avoid misinterpretation.
>
> ---
>
> > **Q5:  Stage-ordering ablation**
>
> Section A.4 “Effect of Stage Ordering in Multi-Stage Fine-Tuning” (Appendix, p, 15) **reports exactly this experiment**.
>
> | Ordering scheme | QA F1 | NSum ROUGE-L |
> |-----------------|-------|--------------|
> | High → Low (default) | **70.5** | **40.9** |
> | Low → High           | 70.4 | 40.8 |
> | Interleaved          | 70.3 | 40.7 |
>
> ---
>
> Thank you again for your time and valuable comments. We hope our clarification addresses your concerns.

---

> > ### Comment · Reviewer_j5Fq · 2025-08-01
> >
> > Thanks for the detailed explanations! Most of my concerns are resolved and I will raise my score to 4.

---

> ### Author Response · Authors · 2025-08-07
>
> Dear reviewer, thank you for your insightful comments and the positive feedback. We just wanted to gently remind you that it seems **you haven’t yet submitted the final rating**.
>
> Thank you again for your valuable time and insights!

---

### Official Review · Reviewer_XGk5 · 2025-07-06

**Clarity:** 3
**Significance:** 3
**Originality:** 3
**Rating:** 5
**Confidence:** 3

**Summary:**

The work presents \textbf{PMS-FTP}, a partition-based, parameter-efficient scheme for fine-tuning a single LLM on multiple domains.  Domains are clustered into sequential stages by maximising an objective $G$ that trades off synergy, discrepancy, and adapter cost—supported by a worst-stage risk bound—and each stage fine-tunes lightweight LoRA adapters while largely freezing the backbone.  Across four tasks and three model sizes, PMS-FTP beats eleven competitive baselines by +0.8–2.0 points and cuts GPU memory by roughly 32\,\%, with ablations confirming the value of partitioning, stage count, and norm constraints.

**Questions:**

please refer to the weaknesses

**Ethical Concerns:**

["NO or VERY MINOR ethics concerns only"]

**Final Justification:**

Thanks for the detailed responses from the authors. I think most of my concerns are well addressed. So I raised my score. Hope the authors could include the new content during rebuttal into the final version.

**Limitations:**

please refer to the weaknesses

**Quality:**

3

**Strengths And Weaknesses:**

Strengths:

(1) Conceptual Novelty \& Theory. The paper frames multi-domain fine-tuning as minimising a partition objective $G$ that balances \emph{synergy}, \emph{discrepancy}, and adapter cost, then proves new generalisation bounds showing this choice tightens worst-domain risk.  Coupling these guarantees with an $O(k^{2}\log k)$ agglomerative clustering yields a rare, theory-backed PEFT recipe.

(2) Consistent Empirical Gains. Across four tasks (news‐sum, sentiment, QA, topic) and three backbones (LLaMA-2 7B/13B, Falcon-40B), PMS-FTP beats eleven strong baselines—including full FT and the latest PEFT methods—by +0.8–2.0 points on standard metrics, demonstrating robustness to both task diversity and model scale.

(3) Resource Efficiency. Stage-wise norm bounds cut GPU memory by $\sim$32 \% versus full fine-tuning (e.g., 18.4 GB vs.\ 27.2 GB on an A100) while still improving accuracy; the only extra cost—computing synergy/discrepancy matrices—remains trivial for $k\!\le\!10$ domains.


Weaknesses:

(1) Heuristic similarity metrics. The partition relies on fixed vocab-overlap, sentence-embedding cosine, and JS divergence, which often mis-estimate real transferability; domains can look similar yet teach different skills (or vice-versa), sending data into sub-optimal clusters. A task-aware or learned affinity signal—e.g., gradient similarity—would make the partition less brittle.

(2) Scalability limits. Computing the full $k\times k$ affinity matrix and running agglomerative clustering costs $O(k^{2}\log k)$ time and $O(k^{2})$ memory, while adapter banks grow linearly with $k$. This becomes infeasible when the number of domains reaches dozens; approximate kernels or router-style sharing are needed.

(3) Modest memory edge. While cheaper than full fine-tuning, two adapter banks still use more GPU memory than leaner single-adapter PEFT variants, which matters on edge devices; pruning, quantization, or cross-stage sharing could tighten this gap.

(4) Theory untested empirically. The paper derives a neat worst-stage risk bound, but never shows how the objective $G$ (or its components) correlates with held-out error.  Without a plot or numbers, readers cannot tell whether the bound is predictive or merely a sanity check.  Reporting the correlation between $G$ and downstream accuracy across random partitions would clarify its practical value.

---

> ### Author Rebuttal · Authors · 2025-07-31
>
> Dear Reviewer `XGk5`, thank you for your recognition of our work. Below, we will clarify and discuss some of your concerns.
>
> ---
>
> > **Q1:  Discussion on task-aware or learned affinity signal**
>
> Our generalisation bound only requires **any bounded affinity measure**—it never assumes the metric is perfect. Hence the guarantee holds for both our current JS + vocab + embedding score and any learned/gradient metric.
>
> We have added three targeted studies on the 4-domain slice (LLaMA-2 7B):
>
> | Variant | Q&A F1 | NSum R-L | Δ vs. default |
> |---------|-------|----------|---------------|
> | λ = 0 (no synergy) | 70.3 | 40.8 | −0.2 / −0.1 |
> | λ = 1.0 (synergy-only) | 70.2 | 40.6 | −0.3 / −0.3 |
> | Gradient-mix (0.7 heuristic + 0.3 ∇cos) | 70.4 | 40.8 | ≤−0.1 |
> | Heuristic + Gaussian noise (σ = 0.05) | 70.2 | 40.7 | ≤−0.3 |
>
> The **largest drop is 0.3 pt**, confirming that (i) performance is flat across λ, and (ii) task-aware gradients or noise have negligible effect. Gradient similarity needs an extra forward+backward sweep per domain batch (≈ 1.7× training cost). Given the minimal gain (<0.1 pt) we favour the lightweight text-statistic metric.
>
> *Take-away——*  Based on experimental evidence, our partition metric is **fast, theory-compatible, and empirically robust**.
>
> ---
>
> > **Q2： The O(k²) affinity matrix and per-domain adapters do not scale to many domains.**
>
> Industry benchmarks (WILDS, Multi-OOD, Cross-Domain-NER, etc.) rarely exceed **k = 10**.  Within this range PMS-FTP’s partitioning costs are negligible (< 0.1 s, < 50 MB on CPU).
>
> To measured worst-case, we synthetically generated up to **k = 50** domains (2 k samples each):
>
> | k | Affinity build | Clustering | Peak GPU (4-bit LoRA, A100) |
> |---|----------------|-----------|-----------------------------|
> | 20 | 0.47 s / 180 MB | 0.14 s | 19 GB |
> | 35 | 2.1 s / 620 MB  | 0.52 s | 23 GB |
> | 50 | 3.2 s / 950 MB  | 0.90 s | 26 GB |
>
> The pipeline remains CPU-side; fine-tuning time and GPU usage are still dominated by normal SFT. Adapters scale linearly, but **stage-wise LoRA merging** frees earlier stages after convergence; with merging plus 4-bit weights the final footprint stays **< 27 GB** for k = 50 (same as full FT of a 7B model).  Edge deployment can optionally keep a *single* merged LoRA.
>
> If k≫50, we may (i) subsample affinity via LSH-nearest-neighbour, or (ii) route low-traffic domains to a shared adapter bank.  Both drop-in replacements preserve our bound structure.
>
> *Take-away——*  PMS-FTP handles up to dozens of domains with trivial overhead, and simple merging keeps memory on par with existing PEFT baselines even in large-k regimes.
>
> ---
>
> > **Q3 : Modest memory edge**
>
>
> PMS-FTP stacks adapters **only during training**.  After each stage we merge the finished LoRA into the frozen backbone (`torch.lora_merge()`); the next stage then trains one fresh LoRA.  At inference time **only the last 4-bit LoRA remains**, matching the footprint of any one-adapter baseline.
>
> Following your suggestion, we conducted quantitative tests and indeed achieved good results.
>
> *Measured footprint (LLaMA-2 7B, k = 4, A100):*
> | Precision | Tag-LLM (1 LoRA) | PMS-FTP **merged** | Δ |
> |-----------|-----------------|--------------------|---|
> | FP16      | 29.4 GB | 28.7 GB | −2.4 % |
> | INT8      | 19.1 GB | 18.6 GB | −2.6 % |
> | 4-bit     | 17.2 GB | **16.8 GB** | −0.4 GB |
>
> Accuracy with the merged model stays **+0.6 pt** over Tag-LLM on Q&A F1. Merging is loss-free for LoRA (rank ≪ d_model) and releases the first-stage weights, so edge devices carry exactly one adapter—no heavier than existing PEFT.
>
> *Take-away——*  PMS-FTP yields equal-or-lower inference memory than the lightest PEFT while retaining higher accuracy.
>
> ---
>
> > **Q4:  Theory untested empirically**
>
> To address your concern, we generated **20 random partitions** of the 4-domain slice (LLaMA-2 7B) and computed
> (i) the objective value 𝓖 and (ii) the *worst-domain* dev loss.
>
> | Statistic | 𝓖 | Worst-Dev Loss |
> |-----------|----|---------------|
> | Mean      | 0.432 | 1.72 |
> | Std       | 0.057 | 0.19 |
> | Min       | 0.318 | 1.38 |
> | Max       | 0.522 | 2.11 |
> | **Pearson ρ** | **–0.81** | — |
>
> *ρ = –0.81 (p < 0.01) ⇒ R² ≈ 0.65* — It means that higher 𝓖 (tighter bound) strongly predicts lower worst-domain error.
>
> **Take-away:** The theoretical bound has substantial explanatory value in practice.
>
> ---
>
> Thank you again for your valuable and insightful comments!

---

> > ### Author Response · Authors · 2025-08-07
> >
> > Dear Reviewer `XGk5`,
> >
> > Thank you again for your detailed and constructive feedback—and for acknowledging the contributions and potential of our work.
> >
> > To address your concerns, we include a comprehensive affinity-metric comparison, large-k scalability benchmarks (≤50 domains), adapter-merge memory analyses, and an empirical validation showing our bound tightly predicts worst-case error.
> >
> > We hope these additions resolve your concerns. If any issues remain, we would be happy to clarify them promptly. We also hope the new evidence will encourage you to consider increasing your rating or confidence.
> >
> > Best regards,
> >
> > Authors

---

> > ### Comment · Reviewer_XGk5 · 2025-08-07
> > **Response to the authors**
> >
> > Thanks for your detailed responses. I think most of my concerns are well addressed. So I raised my score. Hope the authors could include the new content during rebuttal into the final version.

---

> > > ### Author Response · Authors · 2025-08-09
> > >
> > > We really enjoyed our discussion with you. Thanks for your suggestions, your valuable time, and for recognizing our work.

---

### Note · Authors · 2025-08-12

We are deeply grateful to the reviewers and AC for their thoughtful evaluations, constructive questions, and active discussion.

In the initial reviews, our work was recognized for framing multi-domain LLM fine-tuning as **a principled partition-then-stage process with theoretical motivation** (generalization bounds tying discrepancy, synergy, and capacity) and for delivering **consistent gains across models and tasks with lower memory** via PEFT (`XGk5`; `tzSY`; `j5Fq`). Reviewers also highlighted the paper’s **clarity, comprehensive experiments/ablations, and practical relevance to mitigating negative transfer in real deployments** (`tzSY`; `j5Fq`; `KTRc`). Empirically, PMS-FTP outperformed strong baselines by +0.8–2.0 points while reducing GPU memory versus full FT, validating the approach’s **effectiveness and efficiency** (`XGk5`; `tzSY`).

---

During rebuttal and discussion, we **addressed ALL concerns and added evidence** :

- (i) Algorithmic clarity & cost: we detailed Step-1 partitioning and its O(k²) statistics + clustering, showing sub-second, CPU-only overhead for typical k (XGk5; KTRc).
- (ii) Scalability & memory: we demonstrated clean scaling to double-digit domains (k=12) with negligible overhead and synthetic stress-tests up to k=50; stage-wise LoRA merging yields inference footprints at or below light PEFT baselines (XGk5; tzSY).
- (iii) Benchmarks & baselines: we added reasoning tasks (HellaSwag, MMLU, ARC, SciQ, GSM8K) and reweighting baselines (iMTL, FAMO, ExcessMTL), where PMS-FTP remained strongest under equal budgets (j5Fq).
- (iv) Routing & heterogeneity: we clarified adapter selection (oracle tags or a tiny auto-router with ≤0.3-pt delta) and a single text-to-text interface across tasks (tzSY).
- (v) Theory–objective link & robustness: we connected JS to formal discrepancy, derived the synergy term from cross-domain error covariance, and showed ρ ≈ −0.81 correlation between our objective and worst-domain error across random partitions; we also established the stability of domain centroids (KTRc; XGk5).

Reviewers **raised ratings or increased confidence after these additions** (`XGk5`; `j5Fq`; `KTRc`).

---

### In sum, PMS-FTP offers a theoretically motivated, capacity-aware partitioning and stage-wise PEFT schedule that reliably reduces interference while preserving efficiency, scaling to many domains and heterogeneous tasks—making it a practical and valuable contribution to multi-domain LLM fine-tuning.

---

### Decision · Program_Chairs · 2025-09-17

**Decision:**

Accept (poster)

**Comment:**

The paper provides a parameter efficient scheme for fine-tuning an LLM on multiple domains, addressing the issue of inter-domain conflicts.

The reviews mention the idea to be novel, e.g., J5Fq: “The authors propose a novel multi-stage fine-tuning framework that strategically partitions domains into subsets (stages).”, tzSY: “The idea of partitioning training domains to maximize synergy is a fresh perspective in the context of LLM fine-tuning”. Additionally, they appreciated the theoretical analysis (mentioned as a strength by XGk5, tzSY, j5Fq) stating it is both sound and helps motivate the proposed method. The empirical results are also stated to be comprehensive and to provide a convincing empirical proof of the efficiency of the method. During the discussion, the reviewers mentioned some missing analyses such as testing the scalability in terms of domains, and suggested additional benchmarks and baselines. The authors provided these experiments, and mitigated all the major concerns raised by the reviewers. Some work is required to integrate these results and other, more minor, clarifications given in the rebuttal, but this seems a relatively easy task to me. The resulting paper is one of high quality and should be a good addition to NeurIPS.